# Impact of the Indian Ocean Sea Surface Temperature on the Southern Hemisphere Middle Atmosphere

Chengyun Yang[1,2,3], Xiang Guo[1], Tao Li[1,2,3*], Xinyue Wang[4], Jun Zhang[5], Xin Fang[1,2,3] and Xianghui Xue[1,2,3]

[1] School of Earth and Space Sciences, University of Science and Technology of China, Hefei, 230026, China

[2] CAS Center for Excellence in Comparative Planetology/CAS Key Laboratory of Geospace Environment/Mengcheng National Geophysical Observatory, University of Science and Technology of China, Hefei 230026, China

[3] Hefei National Laboratory, University of Science and Technology of China, Hefei 230088. China

[4] Department of Atmospheric and Oceanic Sciences, University of Colorado, Boulder, U.S.

[5] Atmospheric Chemistry Observations & Modeling Laboratory, NSF National Center for Atmospheric Research

*Correspondence to*: Tao Li (litao@ustc.edu.cn)

**Abstract.**

An index representing the midlatitude Indian Ocean Dipole (MIOD) is derived from the second empirical orthogonal function (EOF) mode of austral winter (JJA) sea surface

temperature (SST) anomalies to examine its impact on the Southern Hemisphere middle and upper atmosphere. ERA5 reanalysis datasets, together with satellite observations and WACCM6 simulations, are used to analyze the associated atmospheric responses to MIOD. The results show pronounced but asymmetric impacts between positive and negative events, primarily driven by differences in wavenumber-1 planetary wave activity. Positive MIOD events enhance

planetary-wave propagation from the Indian Ocean sector, leading to momentum deposition, variations in temperature, zonal winds, as well as a strengthening of the residual meridional circulation. These dynamical anomalies warm the midlatitudes and modify the vortex's vertical–meridional structure. Changes in stratospheric winds further regulate gravity-wave filtering,

providing a pathway for SST signals to extend upward into the mesosphere. In contrast, negative
events generally produce weaker and less statistically robust signals. These results identify
Indian Ocean SST variability as an additional driver of large-scale atmospheric variability and
reveal a previously underappreciated pathway through which the Indian Ocean can influence the
middle and upper atmosphere at interannual timescales.

**1 Introduction**

The stratosphere plays a crucial role in atmospheric vertical coupling and in shaping the
evolution of the climate system, through both radiative and dynamical processes that connect it
to the troposphere and influence global climate variability. Stratospheric processes-including
thermal radiation and radiative–chemical–dynamical coupling-have been shown to influence
both tropical and extratropical circulation, with further effects on surface temperature **(Joshi et
al., 2006; Maycock et al., 2013; Shindell, 2001; Solomon et al., 2010; Tandon et al., 2011**).
Dynamically, extratropical stratospheric anomalies, particularly variations in the polar vortex,
can propagate downward within 1–2 months and significantly disturb tropospheric circulation
**(Baldwin and Dunkerton, 2001; Limpasuvan et al., 2005; Mitchell et al., 1999)**.
In the Southern Hemisphere (SH), such stratospheric disturbances play an especially
prominent role in vertical coupling across the troposphere, stratosphere, and mesosphere, while
also interacting with the tropospheric climate system. For instance, the major SSW events in the
SH during the 21st century triggered vertically coupled atmospheric responses dominated by
wave-mean flow interaction dynamics **(Dowdy et al., 2004; Mbatha et al., 2010; Yang et al.,
2022)**. The variation in the troposphere and lower stratosphere can also affect the upper
atmosphere through modulating wave-mean flow interaction **(Black and McDaniel, 2007;
García-Herrera et al., 2006; Li et al., 2013, 2016a; Yang et al., 2017; Karlsson et al., 2011;
Lindzen, 1981)**.
In addition to internally generated variability, external forcing from the ocean also exert
important influences on stratospheric processes. Tropical sea surface temperature (SST)
variations, particularly those associated with ENSO, are known to influence stratospheric
processes through both water vapor transport and dynamical pathways such as Rossby wave
propagation **(Domeisen et al., 2019; Garfinkel et al., 2013, 2021; Li et al., 2016a; Yang et al.,**

**2015)**. Beyond the tropical Pacific Ocean, SST anomalies in other ocean basins—including the

Indian and Atlantic Oceans-have also been shown to affect the stratosphere **(Xie et al., 2018; Zhou et al., 2018)**. For example, variability in the Indo-Pacific warm pool has been associated with changes in lower stratospheric humidity, with warming leading to drying and cooling to moistening **(Xie et al., 2018)**. Long-term warming of the Atlantic Ocean has contributed to the observed increase in stratospheric water vapor over the past century **(Xie et al., 2020)** and can

affect stratospheric circulation at mid- and high latitudes by inducing atmospheric teleconnections and modulating planetary wave propagation **(Rao and Ren, 2018)**.

Among these oceanic influences, variability within the tropical Indian Ocean has distinct internal modes that significantly modulate tropospheric and stratospheric circulation. The Indian Ocean Basin Mode (IOBM), characterized by basin-wide SST warming, affects precipitation in

the troposphere **(Xie et al., 2009)** and influences the stratosphere by altering planetary wave propagation **(Rao and Ren, 2016; Li et al., 2016b)** and the strength and position of the polar vortex **(Rao and Ren, 2020)**. It has also been linked to the Northern Annular Mode (NAM) response that can offset the effects of tropical Pacific SST anomalies **(Fletcher and Kushner, 2011)**. The Indian Ocean Dipole (IOD), defined by a zonal SST gradient, influences regional and

remote climate by modulating large-scale circulations and rainfall variability **(Ashok et al., 2001; Guan and Yamagata, 2003; Ramsay et al., 2017; Saji et al., 1999; Tozuka et al., 2016)**. Beyond its tropospheric impacts, the IOD has also been identified as a driver of Southern Hemisphere climate variability and may affect the stratosphere through the excitation of extratropical wave trains and modulation of wave propagation **(Saji and Yamagata, 2003;**

**Risbey et al., 2009; Son et al., 2013;  Rao et al., 2020)**. Strong IOD events in particular have been linked to persistent annular mode responses and stratospheric teleconnections **(Fletcher and Cassou, 2015; Bègue et al., 2010)**.

While extensive efforts have been made to understand the impacts of tropical Indian Ocean variability-including both basin-wide warming (IOBM) and zonal SST gradients (IOD)-on

tropospheric and stratospheric circulation, relatively little attention has been given to SST anomalies outside the tropical region. Distinctive SST variability modes also exist in the extratropical Southern Hemisphere Indian Ocean, including the Subtropical Indian Ocean Dipole (SIOD) **(Behera et al., 2000; Swadhin and Yamagata, 2001)** and Ningaloo Niño **(Ryan et al., 2021)**, each characterized by unique spatial and temporal patterns. However, the influence of

these subtropical Indian Ocean SST indices on the middle and upper atmosphere remains largely

unexplored, despite their potential importance in modulating stratospheric dynamics through planetary wave excitations.

Considering the substantial hemispheric asymmetry in ocean coverage-the SH is significantly more ocean-dominated (81%) compared to the Northern Hemisphere (NH, 61%) **(Smith and Sandwell, 1997)**, planetary wave activity in the Southern Hemisphere is predominantly influenced by oceanic thermal forcing rather than orographic effects **(Brayshaw et al., 2008; Held et al., 2002)**. Over 90% of the net increase in global ocean heat content in recent decades has occurred in the Southern Hemisphere **(Rathore et al., 2020)**, highlighting the growing climatic influence of Southern Hemisphere oceanic processes. These considerations raise the possibility that SST variability over the subtropical to midlatitude Indian Ocean may exert a more direct and underappreciated influence on stratospheric circulation via modulation of planetary wave activity.

Since conventional Indian Ocean SST indices predominantly represent SST variability peaking during austral summer, coincident with the peak of ENSO, their utility in characterizing the austral winter ocean–atmosphere coupling remains limited. Yet the atmospheric background conditions during austral winter are more favorable for planetary wave propagation into the stratosphere, thereby promoting vertical atmospheric coupling **(Charney and Drazin, 1961)**. Considering recent changes in the spatiotemporal characteristics of Indian Ocean SST anomalies under climate change, this study aims to re-examine the wintertime SST variability from the tropical to midlatitude Indian Ocean during the Austral wintertime (June-July-August, JJA) and investigate its impacts on the Southern Hemisphere stratosphere and higher atmospheric layers. The remainder of this paper is organized as follows. Section 2 describes the datasets, model simulations, and analysis methods used in this study. Section 3 presents the identification of the SST variability modes, the definition of event years, and the associated stratospheric responses. Section 4 examines how the influences extend further into the mesosphere, analyzing the relevant dynamical processes and discussing their implications for the interpretation of the results. Section 5 provides a summary of the main findings.

## 2 Data and Method

### 2.1 Observations and simulations

Previous studies have shown that major Indian Ocean SST modes, such as the subtropical Indian dipole, typically peak during the Northern Hemisphere winter **(e.g. Behera et al., 2000;**

**Swadhin and Yamagata, 2001)**. Considering the emerging spatiotemporal changes in Indian Ocean SST patterns under ongoing climate change **(e.g., Tong et al., 2025)**, we re-evaluate the dominant variability of Indian Ocean SST anomalies during the austral winter (JJA) using
Empirical Orthogonal Function (EOF) analysis. The Hadley Centre Sea Ice and Sea Surface Temperature (HadISST) dataset, developed by the UK Met Office, was used to analyse the spatiotemporal characteristics of SST variability over the midlatitudes of the Indian Ocean **(Rayner et al., 2003)**. HadISST provides a long-term, globally complete record of sea surface temperature and sea ice concentration based on merged observations from ships, buoys, and
satellites, with quality control and homogenization applied. Monthly fields at $1° \times 1°$ resolution were used for 1980–2022, from which JJA means were calculated for analysis.

To investigate the atmospheric response to the SST anomalies, this study uses atmospheric temperature, wind, and geopotential height (hgt) data from the European Centre for Medium-Range Weather Forecasts (ECMWF) Reanalysis v5 (ERA5). ERA5 is the fifth-generation global
reanalysis dataset developed by ECMWF, produced using advanced data assimilation techniques and the IFS Cy41r2 forecasting model **(Hersbach et al., 2020)**. It combines a wide range of observations, including satellite, radiosonde, and surface station data, to provide high-resolution, spatiotemporally continuous climate fields. The variables used in this study are daily means for the period 1979-2022, extracted at $1° \times 1°$ horizontal resolution on 37 standard pressure levels
extending up to 1 hPa (~50 km), providing adequate coverage of the stratosphere.

This study also uses temperature observations from the SABER and Halogen Occultation Experiment (HALOE) satellite instruments **(Russell et al., 1993)**. The HALOE onboard the Upper Atmosphere Research Satellite (UARS), operated from October 1991 to November 2005 as part of NASA's Mission to Planet Earth. HALOE retrieved vertical profiles of temperature and several
trace gases (e.g., $O_3$, HCl) by solar occultation, using multiple infrared channels. Temperature profiles in the middle atmosphere (~40-80 km) were derived from $CO_2$ absorption near 2.8 μm, with a vertical resolution of approximately 4 km and an accuracy of ~5 K below 80 km. HALOE provided daily sunrise and sunset observations within ±50° latitude, though data are sparse in the summer hemisphere.

The Sounding of the Atmosphere using Broadband Emission Radiometry (SABER) instrument aboard NASA's Thermosphere Ionosphere Mesosphere Energetics and Dynamics (TIMED) satellite has been operating since December 2001 **(Russell Iii et al., 1999)**. It alternates between northern (52°S–83°N) and southern (83°S–52°N) hemisphere viewing modes, providing

full local time coverage approximately every 60 days. SABER measures limb radiance in the 15 μm $CO_2$ band to retrieve vertical temperature profiles from ~10 to 120 km, covering the stratosphere, mesosphere, and lower thermosphere. The vertical resolution is ~2 km, with an accuracy better than 2 K in the stratosphere and mesosphere. The SABER Version 2.0 temperature dataset offers long-term, high-quality observations widely used for studying atmospheric waves and variability.

Simulations from the Specified Dynamics configuration of Whole Atmosphere Community Climate Model version 6 (WACCM6-SD) were used to examine the upper atmospheric response to Indian Ocean SST anomalies. WACCM6 is a high-top atmospheric model developed at NCAR as part of CESM2, with comprehensive physical and chemical processes extending from the surface to the lower thermosphere **(Gettelman et al., 2019)**. In the SD configuration, meteorological fields are nudged toward MERRA-2 reanalysis every six hours to reduce internal variability and model bias. WACCM6 is nudged toward MERRA-2 below approximately 0.1 hPa (~50–60 km), with a smooth tapering of the relaxation coefficient near the upper boundary of the nudged region. Above this altitude, including the mesosphere and lower thermosphere, the model evolves freely. This setup allows the stratospheric variability to follow the reanalysis while retaining internally generated dynamics in the mesospheric region. The version used here was run at $0.95° \times 1.25°$ horizontal resolution with 70 vertical levels and a model top near 140 km. It includes the TSMLT chemistry mechanism, and simulations were conducted for 1980–2022, constrained by historical SST and anthropogenic emissions from the Community Emissions Data System (CEDS).

A multi-source satellite ozone dataset from NASA's Goddard Space Flight Center was used to analyse Antarctic ozone conditions, including total column ozone, ozone hole area, and ozone mass deficit, covering the period from 1979 to the present **(Van Der A et al., 2010)**. The dataset integrates observations from multiple satellite instruments: TOMS onboard Nimbus-7 (1979–1992), TOMS on Meteor-3 (1993–1994), Earth Probe TOMS (1996–October 2004), OMI on Aura (November 2004–June 2016), and OMPS on Suomi NPP (since July 2016), with additional correction and gap-filling using MERRA-2 and GEOS FP data.

### 2.2 Analysis method

To obtain a longer-term record of upper atmospheric temperature, monthly mean data from HALOE and SABER were merged to generate a continuous dataset spanning the period from 1991 to 2020. Temperature profiles from both instruments were first interpolated onto a uniform

grid with 1-km vertical spacing between 40 and 80 km, and 5° latitude intervals between 55°S and 55°N. For HALOE, daily sunrise and sunset profiles were first averaged separately by month, then combined to reduce diurnal tide influences. For SABER, ascending and descending profiles were first averaged daily, and then a two-month running mean was applied to further suppress tidal signals. Due to differences in sampling characteristics, SABER more effectively removes semidiurnal tides, while residual tidal contamination may remain in HALOE monthly means, potentially introducing temperature biases. To construct a unified temperature dataset, HALOE and SABER monthly mean profiles were first bias-corrected based on their mean differences during the overlap period. A constant offset was applied to each dataset to minimize discontinuities, ensuring consistency in the merged time series. The corrected HALOE and SABER records were then concatenated to produce a continuous monthly temperature dataset for the period 1991-2020, following the method described by **Li et al. (2021)**.

Since middle and upper atmospheric variability is influenced by external drivers such as solar activity and the quasi-biennial oscillation (QBO) **(Li et al., 2013)**, a multiple linear regression (MLR) approach was applied to remove their linear effects and long-term trends, thereby isolating the response to Indian Ocean SST anomalies. Following the method of **(Li et al., 2016b)**, monthly anomalies were calculated by subtracting the multi-year monthly climatology from the original data to eliminate the seasonal cycle. The resulting anomalies were then regressed onto several external drivers, including El Niño–Southern Oscillation (ENSO), the QBO, the equivalent effective stratospheric chlorine (EESC), and the 11-year solar cycle. ENSO was represented by the Niño 3.4 index, defined as the 3-month running mean of SST anomalies averaged over 5°N–5°S and 120°–170°W. QBO was captured using two orthogonal predictors: the equatorial zonal-mean zonal wind at 10 hPa (QBO10) and 30 hPa (QBO30) derived from ERA5 reanalysis datasets. The equivalent effective stratospheric chlorine time series used in this study corresponds to the WMO A1-2010 scenario and uses the method suggested by **(Newman et al., 2007)**. 11 year solar activity was represented by the 10.7 cm solar radio flux (F10.7), which serves as a proxy for solar ultraviolet (UV) variability and has been continuously monitored since 1947 by Canadian agencies.

The multiple linear regression model is expressed as follows:

$$T(t)=\alpha \cdot NINO3.4+\beta 1 \cdot QBO10+\beta 2 \cdot QBO30+\gamma \cdot F10.7+\delta \cdot EESC+\kappa \cdot trend+\varepsilon \qquad (1)$$

where $T(t)$ denotes the monthly anomaly of the target atmospheric variable, the term trend represents the long-term linear trend, and $\varepsilon$ is the regression residual, i.e., the atmospheric

component with the influence of unrelated factors removed. The coefficients $\alpha$, $\beta 1$, $\beta 2$, $\gamma$ and $\delta$ correspond to the contributions from ENSO, QBO at 10 hPa and 30 hPa, 11-yr solar cycle and

EESC, respectively.

Given the limited number of events, the Monte Carlo method was employed to assess the statistical significance of the composite results in this work. Specifically, 1,000 synthetic composites were generated by randomly selecting the same number of years as the target events (without replacement) from the whole time period, assuming a null hypothesis that the composite

signal arises purely from internal variability. For each realization, the composite was recalculated to build an empirical distribution. Two-sided significance thresholds were then determined based on the 2.5th and 97.5th percentiles of this distribution (corresponding to a 95% confidence level). If the actual composite value falls outside this range, the null hypothesis is rejected and the result is considered statistically significant.

To quantify the influence of planetary wave-induced momentum and heat fluxes on the zonal-mean circulation, the Eliassen-Palm (E-P) flux and its divergence were calculated following the formulation of **(Andrews et al., 1987)**. The meridional and vertical components of the E-P flux in pressure coordinates are given by:

$$F_y = \rho_0 a \cos\varphi \left( -[u'v'] + \frac{[u]_z[v'\theta']}{[\theta_z]} \right), \qquad (2)$$

$$F_z = \rho_0 a\cos\varphi \left( \left[ f - \frac{(\bar{u}\cos\varphi)_\varphi}{a\cos\varphi} \right] \frac{[v'\theta']}{[\theta_z]} - [u'w'] \right), \quad (3)$$

$$\mathrm{Div} = \frac{1}{a\cos\varphi} \frac{\partial(F_y \cos\varphi)}{\partial\varphi} + \frac{\partial F_z}{\partial z}. \qquad (4)$$

where $\rho_0$ denotes air density, a is the Earth's radius, $\varphi$ is latitude, and f is the Coriolis parameter. The variables $u'$, $v'$, $w'$, and $\theta'$ represent the zonal anomalies of zonal wind, meridional wind, vertical velocity, and potential temperature associated with planetary waves, respectively. Square brackets indicate the zonal mean.

To separate wave-induced disturbances from the mean circulation, the residual mean meridional circulation was derived based on the transformed Eulerian mean (TEM) framework proposed by **(Andrews and McIntyre, 1976)**. By incorporating corrections from the eddy heat flux ($[v'\theta']$), this approach provides a clearer depiction of wave–mean flow interactions. The residual circulation components are given by:

$$v^* = [v] - \frac{1}{\rho_0}\frac{\partial}{\partial z}\left(\frac{[v'\theta']}{\partial\bar{\theta}/\partial z}\right) \tag{5}$$

$$w^* = [w] + \frac{1}{a\cos\phi}\frac{\partial}{\partial\phi}\left(\frac{\cos\phi\,[v'\theta']}{\partial[\theta]/\partial z}\right) \tag{6}$$

To diagnose the dynamical contribution of the residual circulation to ozone variability, we compute meridional and vertical transport terms based on anomalous TEM velocities acting on the climatological ozone gradients. For each JJA month, meridional and vertical components of the residual circulation $(v^{*\prime}, w^{*\prime})$ are obtained by subtracting the monthly climatology from TEM fields, and the anomalous transport tendencies are approximated as

$$M_{phi} = -v^{*\prime}\frac{\partial\chi}{\partial y} = -v^{*\prime}\frac{1}{a}\frac{\partial\chi}{\partial\phi} \tag{7}$$

$$M_z = -w^{*\prime}\frac{\partial\chi}{\partial z} = -w^{*\prime}\frac{1}{H}\frac{\partial\chi}{\partial lnp} \tag{8}$$

Where $\chi$ is the climatological ozone mixing ratio. The latitudinal gradient is computed on a spherical coordinate, and the vertical gradient is evaluated in log-pressure coordinates, with a scale height H = 7 km. The diagnostic in this study isolates the effect of circulation anomalies on ozone without solving the full TEM tracer budget, providing a physically interpretable measure of ozone transport variability relevant to MIOD-related circulation changes.

The Takaya–Nakamura (TN) wave-activity flux **(Takaya and Nakamura, 2001)** is used to diagnose the propagation pathways of stationary and quasi-stationary planetary waves in a zonally varying basic flow. The meridional and vertical components of the flux are derived from the geopotential field and the background zonal-mean zonal wind. In compact form, the three-dimensional flux $W = (F_x, F_y, F_z)$ can be written as

$$\mathbf{W} = \begin{pmatrix} F_x \\ F_y \\ F_z \end{pmatrix} = \frac{p\cos\phi}{2|\mathbf{U}|}\begin{pmatrix} \frac{U}{a^2\cos^2\phi}\left[\left(\frac{\partial\psi'}{\partial\lambda}\right)^2 - \psi'\frac{\partial^2\psi'}{\partial\lambda^2}\right] + \frac{V}{a^2\cos\phi}\left[\frac{\partial\psi'}{\partial\lambda}\frac{\partial\psi'}{\partial\phi} - \psi'\frac{\partial^2\psi'}{\partial\lambda\partial\phi}\right] \\ \frac{U}{a^2\cos\phi}\left[\frac{\partial\psi'}{\partial\lambda}\frac{\partial\psi'}{\partial\phi} - \psi'\frac{\partial^2\psi'}{\partial\lambda\partial\phi}\right] + \frac{V}{a^2}\left[\left(\frac{\partial\psi'}{\partial\phi}\right)^2 - \psi'\frac{\partial^2\psi'}{\partial\phi^2}\right] \\ \frac{f_0^2}{N^2}\left\{\frac{U}{a\cos\phi}\left[\frac{\partial\psi'}{\partial\lambda}\frac{\partial\psi'}{\partial z} - \psi'\frac{\partial^2\psi'}{\partial\lambda\partial z}\right] + \frac{V}{a}\left[\frac{\partial\psi'}{\partial\phi}\frac{\partial\psi'}{\partial z} - \psi'\frac{\partial^2\psi'}{\partial\phi\partial z}\right]\right\} \end{pmatrix} + \mathbf{C}_U M \tag{9}$$

where p is pressure, a is Earth's radius, $\varphi$ and $\lambda$ are latitude and longitude, $\psi$ is the geostrophic stream function, $U$ and $V$ denote the basic-state zonal and meridional winds, $|U| = \sqrt{u^2 + v^2}$, $f_0$ is a representative Coriolis parameter, and $N$ is the Brunt-Vaisala frequency. The explicit forms of the three components involve combinations of the horizontal and vertical

derivatives of ψ′. The mean-flow correction term $\mathbf{C}_U M$ is neglected, as its contribution is

generally small for the quasi-stationary waves considered here.

The TN flux provides a dynamically consistent description of how quasi-geostrophic eddies propagate within a three-dimensional background flow. Under Wentzel–Kramers–Brillouin (WKB) assumptions, its orientation aligns with the local group velocity, while its divergence highlights regions where eddy activity converges or diverges, offering insight into the eddy

forcing on the mean circulation. In this study, the horizontal components $F_x$ and $F_y$ were computed from stream-function anomalies together with the climatological mean winds (1980–2022) obtained from the ERA5 reanalysis.

To diagnose the background flow conditions that favor or inhibit the vertical propagation of large-scale Rossby waves, we compute the quasi-geostrophic refractive index (RI) following the

classical formulation of **(Charney and Drazin, 1961)**. The RI provides a measure of the effective waveguide structure for a specified zonal wavenumber s, and is widely used to assess whether stationary planetary waves can propagate upward from the troposphere into the stratosphere.

The zonal-mean meridional gradient of quasi-geostrophic potential vorticity, $\bar{q}_\phi$,

is written as

$$\bar{q}_\emptyset = 2\,\Omega \cos \emptyset - (\frac{(\bar{u} \cos \emptyset)_\emptyset}{a \cos \emptyset})_\emptyset - \frac{a}{\rho} \left(\frac{f^2}{N^2}\,\rho \bar{u}_z\right)_z \qquad (10)$$

where $\Omega$ is the Earth's rotation rate, $a$ is the Earth's radius, $u$ is the zonal-mean zonal wind, $\rho$ is air density, and $N^2$ is the buoyant frequency ($N^2 = g*\mathrm{d}\ln\theta/\mathrm{d}z$) is the static stability.

The refractive index (RI) for a stationary wave of zonal wavenumber s is then given by

$$RI = \frac{\bar{q}_\emptyset}{a\bar{u}} - \frac{s^2}{a^2 \cos^2 \emptyset} - \frac{f^2}{4N^2 H^2} \qquad (11)$$

where $f = 2\Omega \sin \phi$ is the Coriolis parameter and $H = 7\,km$ is the scale height. Positive RI values indicate regions where vertical propagation is permitted, whereas negative values correspond to evanescent (non-propagating) conditions. In this study, RI is calculated for s = 1 and s = 3 to assess the contrasting waveguide environments relevant to the positive and negative

phases of the MIOD.

**2.3 Middle latitude Indian Dipole events**

JJA-mean SST anomalies relative to the 1980–2020 climatology were used directly in the EOF analysis to characterize interannual SST variability during austral winter. Then, the JJA-mean SST at each grid point was normalized by subtracting its temporal mean and dividing by the standard deviation, to suppress regional variability magnitude and emphasize spatial anomaly structures in the EOF analysis.

An EOF analysis was applied to the standardized JJA-mean SST over the Indian Ocean region (60°S–5°N, 40°E–145°E), yielding the leading two modes of SST variability during austral winter, as shown in **Fig. 1**., EOF1 exhibits coherent SST anomalies from the tropics into the midlatitudes, with opposite-signed anomalies near 40°E–60°E and around 140°E at 45°S. The associated principal component captures the basin-wide warming trend over 1980–2020, and this mode accounts for 21.3% of the variance. The second mode (EOF2) shows a dipole-like pattern, featuring opposite SST anomalies between the region off the west coast of Australia extending into the northeastern tropical Indian Ocean and the central midlatitude Indian Ocean located around 40°S, 70°E. This spatial structure, which explains 13.4% of the total variance, bears some resemblance to the subtropical Indian Ocean Dipole (SIOD), but with a more eastward-displaced configuration. The difference between the JJA EOF2 SSTA pattern and the SIOD pattern may arise from the Indian Ocean response to climate change **(e.g. Sun et al., 2022)**. Notably, the JJA-mean SIOD index, calculated using its traditional definition **(Swadhin and Yamagata, 2001)**, shows a moderate correlation with the PC2 at 0.604. However, distinct differences remain, possibly because the EOF2 pattern is derived solely from JJA SST anomalies and may also reflect evolving spatiotemporal SST characteristics in the Indian Ocean under recent climate change.

To better capture the year-to-year spatiotemporal variability of Indian Ocean SST during austral winter revealed by the EOF analysis, a new SST index was constructed. This index reflects the SST dipole pattern associated with PC2 and is defined as the difference between the area-averaged SST over the eastern margin of the subtropical Indian Ocean (100°–120°E, 5°–40°S) and the central midlatitude Indian Ocean (55°–90°E, 30°–45°S), as indicated by the rectangles in **Fig. 1b**. Using this physically based index rather than PC2 provides a simpler and more intuitive metric for subsequent analyses and avoids the sensitivity of EOF-derived PCs to choices of analysis period and preprocessing. A positive phase of this index corresponds to a pattern characterized by warming in the north-eastern sector and cooling in the central midlatitudes. Since this index reflects the SST gradient between the midlatitudes and the eastern part of the basin, it is referred to here as the Middle-latitude Indian Ocean Dipole (MIOD). As shown in **Fig. 2a**, the temporal

evolution of the MIOD index closely tracks that of the second EOF principal component (PC2), with a correlation coefficient of 0.94.

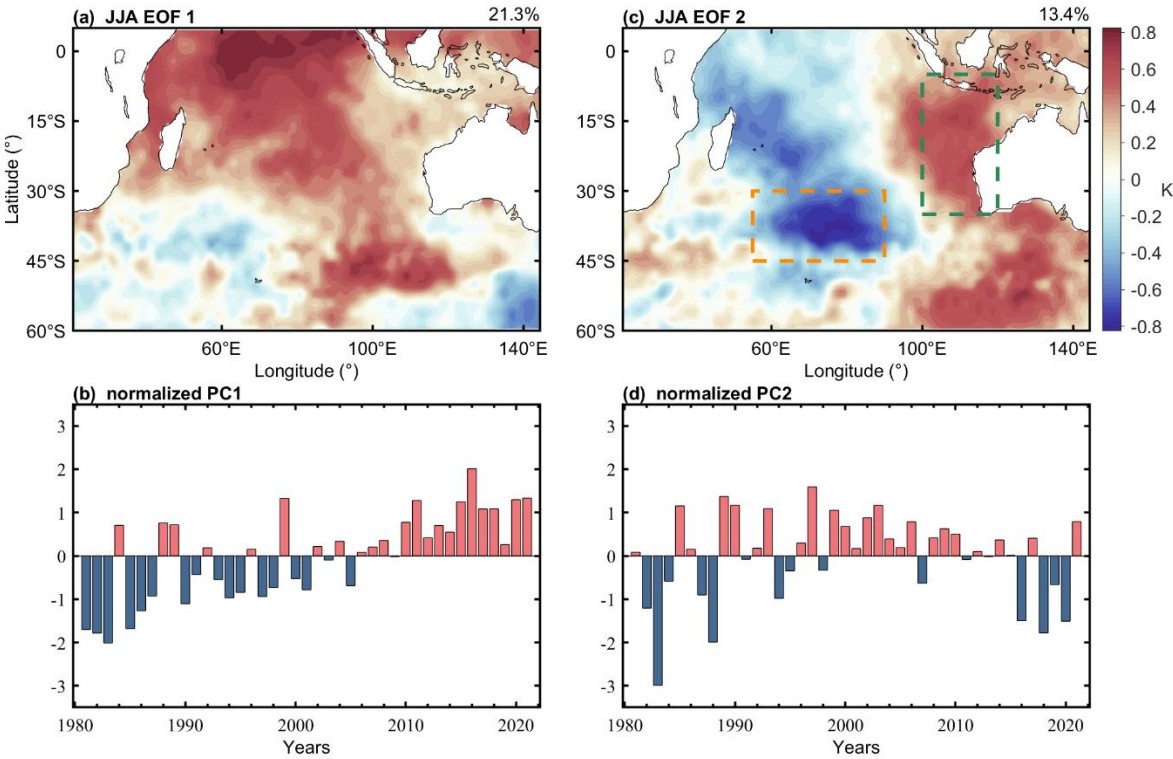

**Figure 1.** SST patterns of (a) EOF1 and (b) EOF2, and principal component time series of (c) EOF1 and (d) EOF2, derived from Indian Ocean SST anomalies during austral winter (JJA) for 1980–2020 over the domain 60°S–5°N, 40°E–145°E.

Based on the standard deviation of the austral winter (June–August) mean MIOD index, years with values exceeding +0.8 standard deviation were defined as positive MIOD events, while those below –0.8 standard deviation were defined as negative events. The MIOD and Niño-3.4 indices show a modest correlation ($r \approx 0.4$), and several MIOD events occur in ENSO years. To ensure the

340 independence of the analyzed samples, ENSO events were identified using the Niño 3.4 index, with a threshold of ±1.0 standard deviation during June–August, and their temporal and spatial overlap, defined as ENSO and MIOD events occurring in the same year, is shown in **Fig. 2**. After excluding overlapping cases, a set of independent MIOD anomaly events was selected, comprising five positive events (1984, 1992, 1996, 2002, and 2005) and seven negative events (1981, 1986,

1993, 2006, 2017, 2018, and 2019).

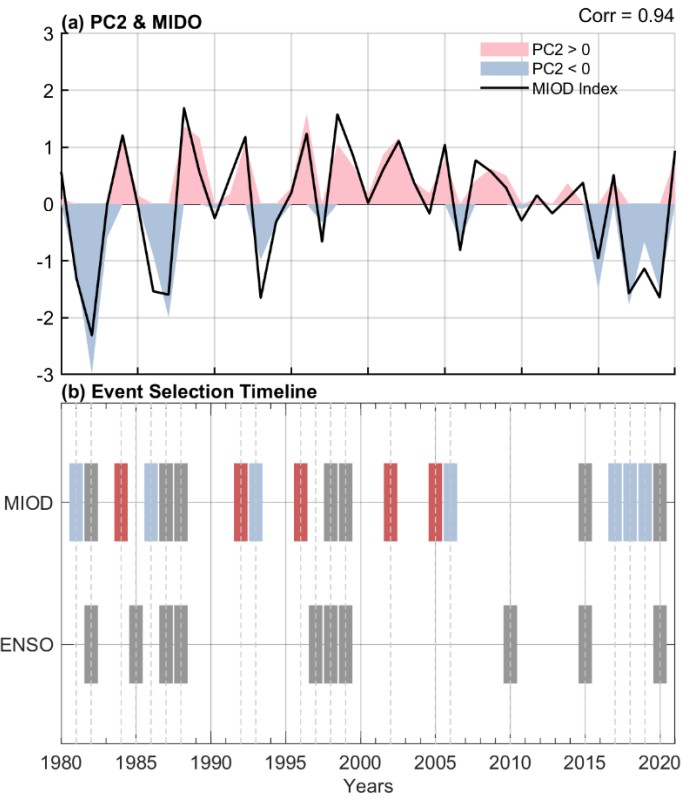

**Figure 2:** (a) Austral winter (JJA) MIOD index (black) and PC2 time series, with positive (negative) PC2 values shaded in red (blue). (b) Two-line timeline summarizing MIOD event selection. Gray bars mark ENSO years. Red and blue bars represent positive and negative MIOD anomalies, respectively, while gray bars in the MIOD row indicate years in which MIOD events overlap with ENSO.

As a robustness check, we also removed the linear influence of ENSO by regressing the MIOD index onto the JJA Niño-3.4 index before identifying events. The resulting MIOD years were nearly identical to those obtained using the threshold-based approach, differing by only one positive event (1991).

## 3 Results

To investigate the impact of the MIOD on the SH stratosphere, composite analyses of zonal-mean temperature and wind fields were performed based on the identified positive and negative MIOD events. **Fig. 3** presents the austral winter (JJA) stratospheric zonal-mean temperature anomalies associated with positive and negative MIOD events. The composites exhibit a pronounced asymmetry between positive and negative MIOD events. During positive

MIOD events (**Fig. 3a** and **Fig. 3b**), a strong negative anomaly in zonal wind appears in the midlatitude stratosphere, centered near 40°S at the stratopause level (~1-3 hPa), with a minimum anomaly exceeding −18 m/s. In contrast, the high-latitude stratosphere exhibits significant positive wind anomalies, centered around 65°S with peak values reaching approximately +8 m/s. These zonal wind anomalies tilt poleward with decreasing altitude, as indicated by the zero-contour shifting from ~50°S near the stratopause (1 hPa) to ~70°S in the lower stratosphere (300 hPa). Over the equator, the upper and middle stratosphere exhibit a vertically alternating pattern, with negative anomalies (–5 m/s) near 3 hPa, positive anomalies (+2 m/s) around 10 hPa, and negative anomalies (~–2 m/s) near 50 hPa. Accompanying these wind changes, significant warming is observed in the Southern Hemisphere stratosphere, forming a "T-shaped" structure. The primary warming appears in the midlatitudes and near the stratopause. A broad region of positive temperature anomalies (> +1 K) extends throughout the stratosphere (300 hPa–1 hPa), intensifying with height and reaching a maximum of over +4 K near 3-10 hPa. The meridional extent also broadens from 45°S–60°S in the lower stratosphere to 30°S–70°S near the stratopause. Additionally, another warming center is detected near the South Pole at ~90°S and 3 hPa, with a peak of +4 K; however, no significant anomalies are present directly below this layer in the mid-stratosphere (> 3 hPa). The zonal wind and temperature anomalies (weakened midlatitude westerlies, strengthened high-latitude westerlies, and polar-cap warming) closely resemble the canonical negative phase of the Southern Annular Mode (SAM). For the negative MIOD events (**Fig. 3c** and **Fig. 3d**), the zonal-mean zonal wind and temperature anomalies are generally weak and fail to reach the 90% confidence level based on the Monte Carlo significance test. Given the limited numbers of positive and negative MIOD events, the composites should be interpreted cautiously. Nevertheless, the overall structures of the atmospheric responses remain internally consistent across the identified events, supporting the robustness of the inferred positive–negative asymmetry within the available record. A brief examination of the adjacent seasons suggests that MIOD-related anomalies outside JJA are generally weak. No statistically significant signals are found in MAM, and only a marginal warm anomaly and slight vortex weakening appear in SON, reaching at most the 90% confidence level. These features imply that the pronounced JJA response shows limited persistence into early spring, underscoring that winter remains the primary season of dynamically robust coupling.

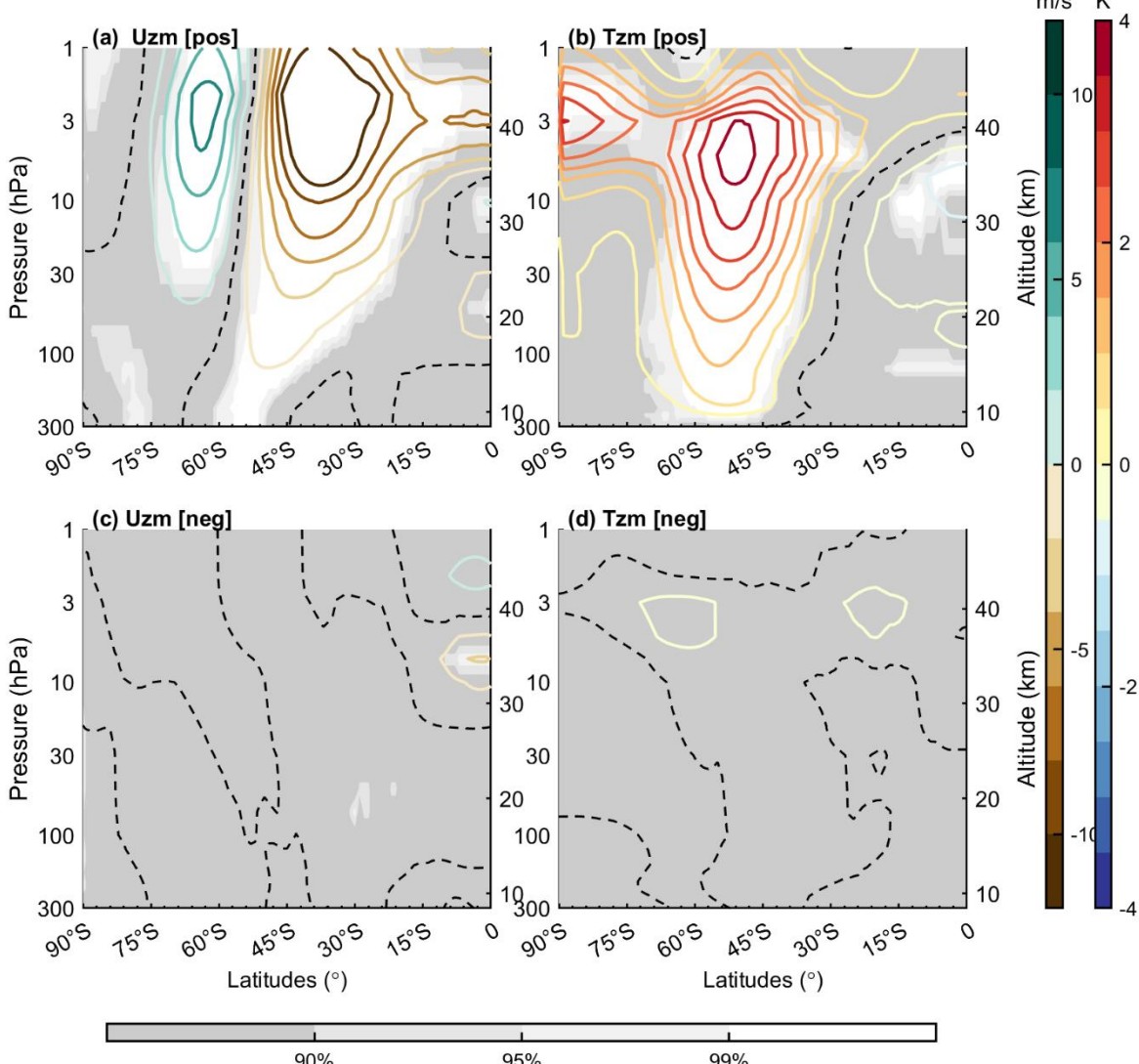

**Figure 3:** (a) Composite zonal-mean zonal wind anomalies for positive MIOD events. Contours represent wind anomalies, with the dashed contour denoting the zero line. Shading indicates statistically significant regions based on a Monte Carlo test. (b) Same as (a), but for the zonal mean temperature anomalies. (c) and (d), similar to (a) and (b) but for negative MIOD events.

This asymmetry between positive and negative events may reflect differences in SST characteristics and atmospheric response mechanisms associated with opposite phases of the MIOD. To explore this further, **Fig. 4** compares the spatial patterns of SST anomalies during positive and negative MIOD events. During positive-phase events (**Fig. 4a**), significant cold anomalies (~ –0.4 K) are centered over the midlatitude central Indian Ocean, extending southeastward from east of Madagascar (30°S, 60°E) to approximately 45°S, 100°E. Concurrent warm anomalies are located south of Australia, with statistically significant values confined to the oceanic region between Australia and Antarctica. In contrast, negative-phase MIOD events

(**Fig. 4b**) are characterized by a pronounced negative SST anomaly (–0.5 K) extending from south of the Maritime Continent toward the western coast of Australia, while warm anomalies are restricted to a relatively narrow region in the midlatitude central Indian Ocean.

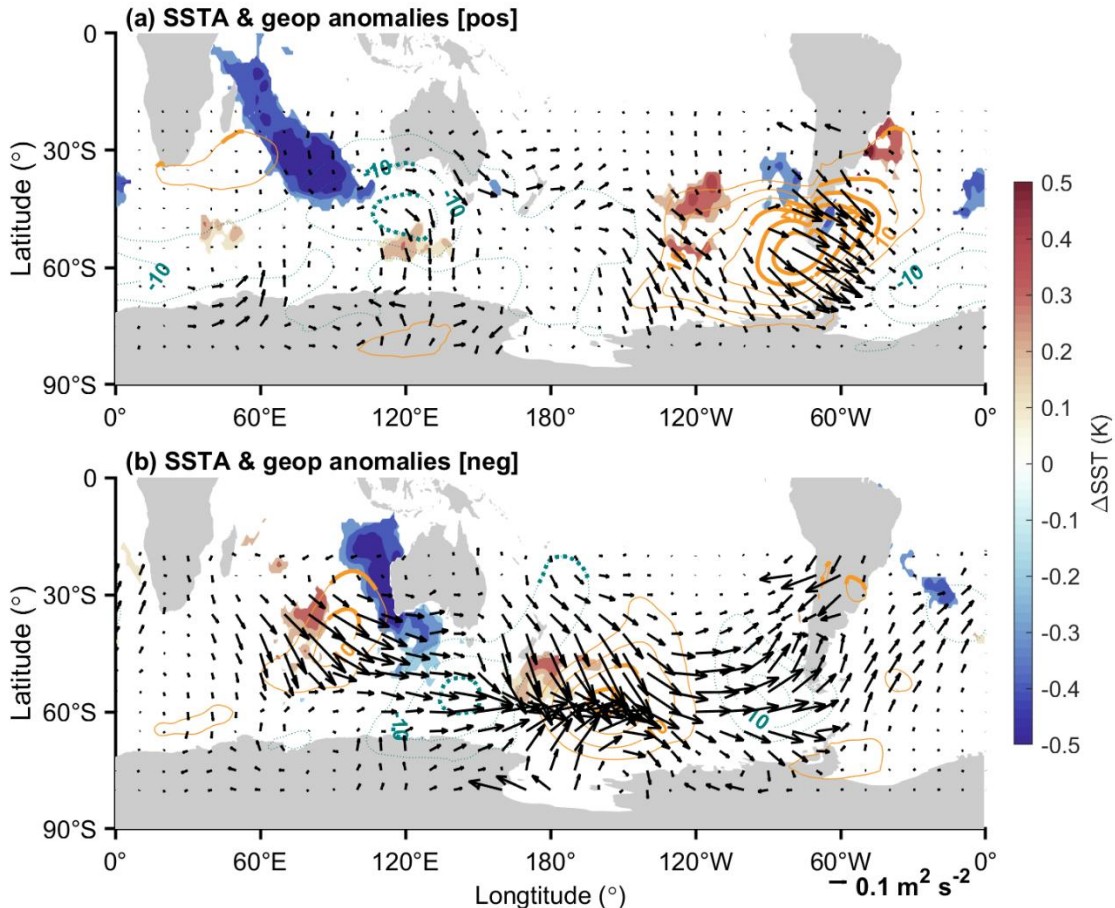

**Figure 4:** Composite anomalies for positive MIOD events during June–August (JJA). (a) Sea surface temperature (SST) anomalies (shading), with only regions passing the 90% Monte Carlo confidence test shown. Overlaid contours indicate zonal anomalous geopotential height at 850 hPa, with orange (blue) lines representing positive (negative) anomalies. Contours are bolded where the anomalies are statistically significant at the 90% confidence level. Black arrows show the Takaya–Nakamura wave activity flux (TN flux) at 850 hPa, illustrating the horizontal propagation of anomalous planetary wave activity. (b) is the 415 same as (a) but for composite of negative events.

Given the ocean-dominated surface characteristics of the SH, planetary wave activity is primarily modulated by thermal forcing associated with zonal SST anomalies and associated diabatic heating. Although both positive and negative phases of the MIOD are characterized by pronounced meridional SST gradients over the Indian Ocean, the asymmetry in their spatial 420 distributions may lead to differences in the atmospheric wave response. Composite anomalies of the zonal deviation of hgt at 850 hPa during the Southern Hemisphere winter (dashed and solid

contours in **Fig. 4a**) indicate that, under positive MIOD conditions, a significant positive center (~+10 m) emerges over the western Indian Ocean to southern Africa (20°–60°E, 25°–45°S), west of the negative SST anomaly at the central Indian Ocean. Meanwhile, large areas of significant

negative anomalies (–10 m) are observed over southern Australia and the southwestern Pacific (90°–140°E, 30°–55°S). Accompanied by a positive hgt anomaly over southern South America and a negative hgt center over the South Atlantic, the composite anomalies form a coherent meridional wave pattern extending across the Southern Hemisphere midlatitudes.

During negative MIOD events (**Fig. 4b**), a localized positive geopotential height anomaly

(+13 m) is located over the southeastern Indian Ocean and the western coast of Australia (90–100°E, 25–45°S). The SH midlatitudes near 60°S exhibit a triple wave pattern, characterized by negative, positive, and negative anomalies centered south of Australia (150°E), over the South Pacific (150°W), and off the west coast of South America (90°W), respectively (−15 m → +15 m → −10 m). These results indicate differences in the 850 hPa geopotential height responses

between positive and negative MIOD events. In particular, positive events are associated with a broader zonal gradient in geopotential height over the southern Indian Ocean, which likely facilitates the excitation of larger-scale and more coherent wave structures.

To access whether the Indian SST anomalies and the global geopotential anomalies are associated, we further examined the TN wave-activity flux at 850 hPa (as indicated by the vector

in **Fig. 4**). For either positive or negative MIOD cases, TN-flux perturbations extend from the midlatitude eastward to the south Pacific, indicating that SST anomalies in either events can modulate the large-scale wave field. The similarity of the TN-flux patterns between positive and negative MIOD events indicates that both phases are capable of exciting planetary-wave activity over the midlatitude Indian Ocean. Thus, the contrasting atmospheric responses between the two

phases are unlikely to arise from differences in the strength or spatial extent of the planetary - wave forcing itself. Instead, this result suggests that the differences in the structure and phase of the associated geopotential height anomalies may motivates a more targeted diagnosis of the planetary-wave characteristics.

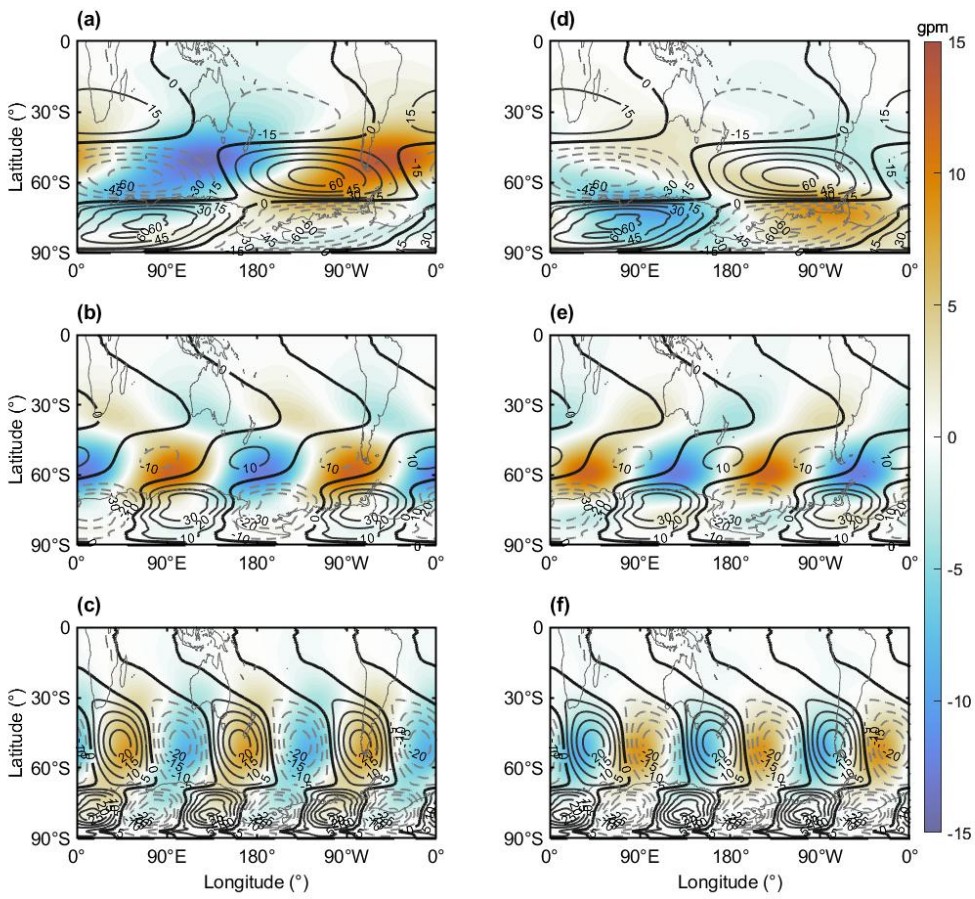

**Figure 5:** Composite anomalies of 850 hPa geopotential height for different zonal wavenumber planetary wave components during Southern Hemisphere winter (June–August, JJA), overlaid with the long-term climatological mean. (a–c) Composite results for positive MIOD events for zonal wavenumbers 1, 2, and 3, respectively; (d–f) same as (a–c), but for negative MIOD events. Shading indicates geopotential height anomalies, and contours represent the climatological mean.

To further diagnose the differences in planetary wave activity modulated by positive and negative MIOD events, a harmonic decomposition was performed on the 850 hPa geopotential height field. Based on zonal Fourier decomposition, the zonal anomalies of the geopotential height at each latitude were expanded and reconstructed to extract the planetary wave components corresponding to zonal wavenumbers 1-3 (WN-1, WN-2, WN-3). This approach allows us to isolate the dominant large-scale wave structures and examine their anomalies relative to the climatological mean, thereby assessing changes in wave phase alignment and amplitude associated with MIOD events. For the dominant planetary wave component, zonal wavenumber 1 (WN-1), the composite anomalies show pronounced differences between positive and negative MIOD events. During positive events (**Fig. 5a**), strong WN-1 anomalies with

amplitudes up to 18 m are excited at midlatitudes, exhibiting an in-phase structure with the climatological mean wave train. This phase alignment enhances the overall WN-1 amplitude by approximately 30%, indicating a substantial reinforcement of the planetary wave. In contrast, during negative events (**Fig. 5d**), weaker WN-1 anomalies (~3 m) emerge over the midlatitudes (40°–60°S), results in a ~5% reduction in wave amplitude. The planetary wave with zonal wavenumber 2 (WN-2) exhibits distinct response characteristics (**Fig. 5b**). Under climatological conditions, the WN-2 amplitude at midlatitudes is relatively weak (~10 m). During positive MIOD events, a similar-magnitude anomaly (~12 m) is generated, but it is nearly out of phase (~180° phase shift) with the climatological wave, substantially reducing the net WN-2 amplitude through destructive interference. Meanwhile, negative MIOD events generate WN-2 anomalies of comparable strength (~10 m), but with a phase structure approximately orthogonal (~90° phase shift) to the climatological pattern. As a result, the anomaly has limited projection onto the background wave and does not significantly alter the overall wave amplitude. The planetary wave component with zonal wavenumber 3 (WN-3) also exhibits distinct responses between positive and negative MIOD events (**Fig. 5c and 5f**). In positive events, the WN-3 anomalies (~10 m) are nearly in phase with the climatological WN-3 pattern, resulting in constructive interference and a ~40% increase in wave amplitude. This reinforces the overall planetary wave activity. Similar to the WN-2 response, the WN-3 anomalies during negative MIOD events (~8 m) exhibit a phase structure that is approximately orthogonal to the climatological mean (~10 m). Consequently, the contribution of the anomaly to the background wave is limited, and the overall amplitude remains largely unchanged.

The differentiated responses of planetary waves with different zonal wavenumbers help explain the contrasting atmospheric responses of positive and negative MIOD events. Positive events, characterized by a unique SST anomaly structure, induce substantial zonal geopotential height gradients over the southern Indian Ocean, which effectively excite WN-1 and WN-3 components. This leads to enhanced planetary wave activity and a longer wave propagation path. On the other hand, negative events exert limited influence on WN-1 and primarily generate WN-2 and WN-3 anomalies that are nearly orthogonal to the climatological waves, resulting in minimal modulation of planetary wave amplitude.

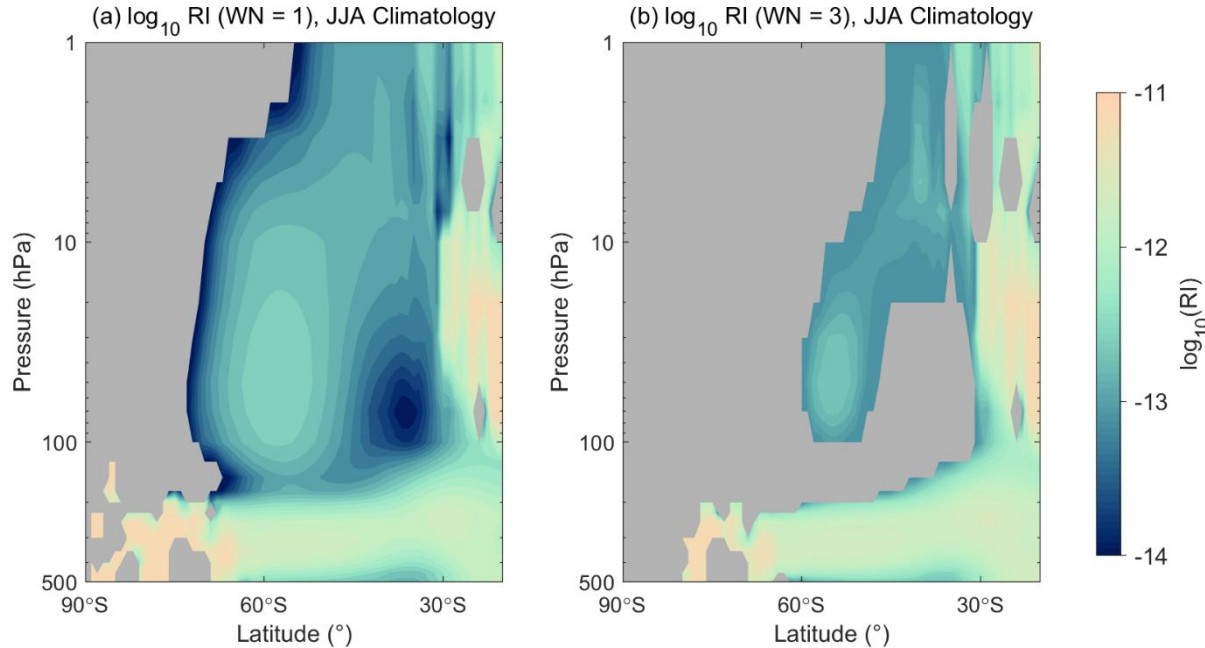


**Figure 6**: June–July–August (JJA) climatological refractive index (RI) for (a) zonal wavenumber-1 (WN-1) and (b) zonal wavenumber-3 (WN-3), averaged over 1979–2020. Negative RI values are masked and shown in gray. The pressure axis is plotted on a logarithmic scale.

In addition to the zonal structure of the wave anomalies, the background waveguide also plays a critical role in determining which wavenumbers can effectively influence the stratosphere. The JJA climatology of the RI reveals a broad, vertically continuous region of positive value for WN-1 extending from the upper troposphere into the mid-stratosphere across the Southern Hemisphere extratropic **(Fig. 6)**, indicating a climatologically open pathway for

large-scale stationary waves. In contrast, WN-3 exhibits a broad region of negative refractive index centered near 100 hPa, extending from Antarctica to approximately 30°S. This extensive negative-RI layer acts as a blocking lid that prevents WN-3 from propagating upward into the stratosphere. This inherent waveguide structure explains why WN-1 dominates the stratospheric response: during positive MIOD events, enhanced WN-1 anomalies can utilize this favorable

propagation environment to penetrate into the stratosphere, whereas WN-3 anomalies-despite showing amplitude changes-remain largely trapped in the lower atmosphere and contribute little to the stratospheric variability.

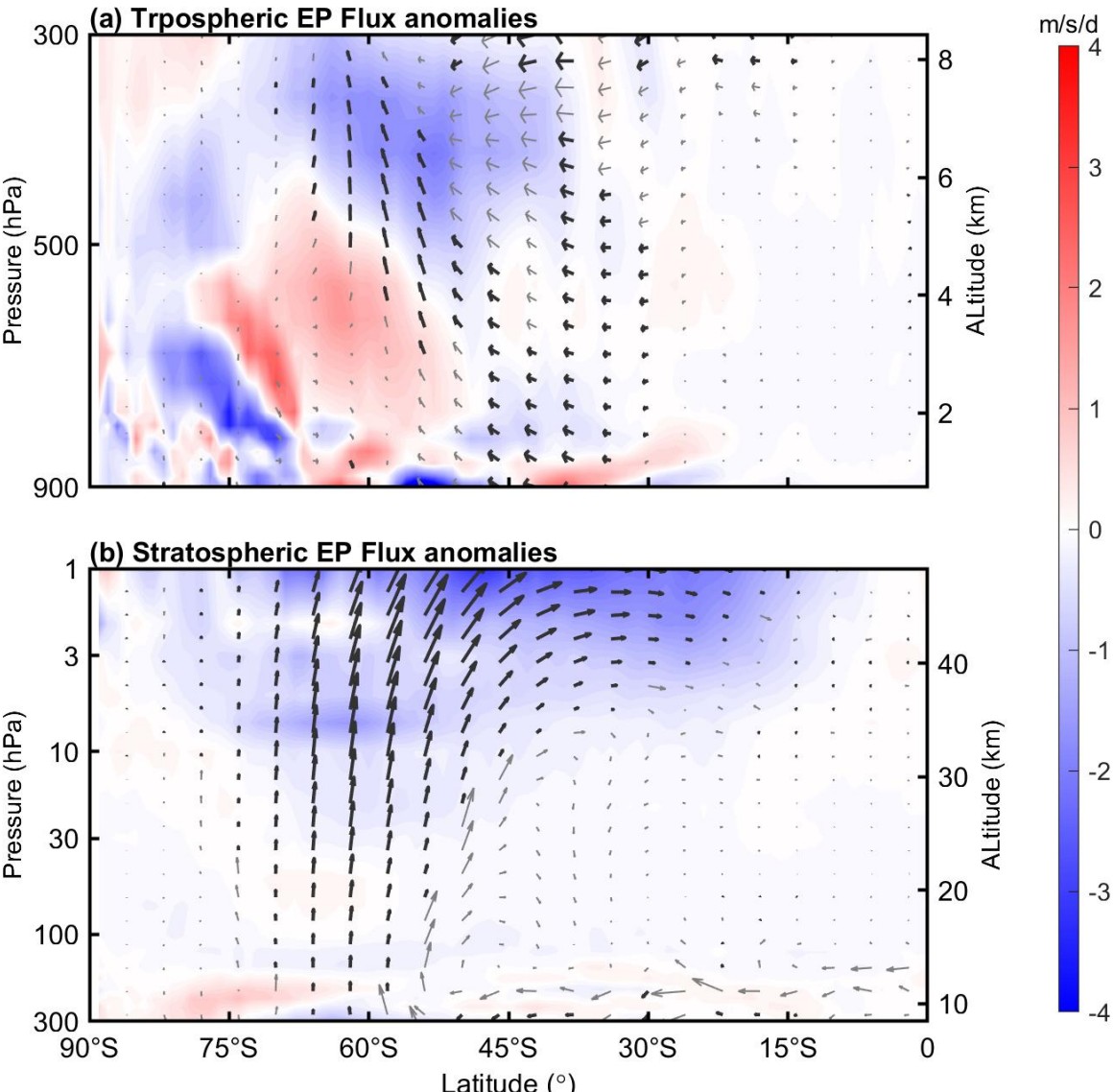

**Figure 7:** (a) Composite anomalies of tropospheric E–P flux components ($F\phi$, $Fz$) during positive MIOD events in June-August (JJA), with $F_\phi$ and $F_z$ normalized by a$\pi$ and $10^6$ respectively. Shading indicates the E-P flux divergence. Arrows denote the anomalous E-P flux vectors, with dark arrows marking vectors that pass the 90% Monte Carlo confidence test. (b) Same as (a), but for the stratosphere, with $F_\phi$ and $F_z$ normalized by a$\pi$ and $10^5$, respectively.

To characterize the resulting vertical propagation of planetary waves, we then examine the E-P flux and its divergence. **Fig. 7** presents the MIOD-related E-P flux anomalies during JJA. For clarity, the tropospheric (900-300 hPa) and stratospheric (300-1 hPa) E-P fluxes are plotted using separate normalization factors to improve the clarity of the plotted vectors since the value in the tropospheric and stratospheric differ substantially in magnitude. The composite results

reveal a clear enhancement of upward planetary wave propagation during positive MIOD events,

consistent with the excitation of stronger WN-1 components.

      In the mid- to upper troposphere (**Fig. 7a**), significant anomalous upward propagation of planetary waves is observed over the midlatitudes (50°S-70°S). These regions of enhanced upward E-P flux coincide well with those identified in the lower stratosphere (**Fig. 7b**), indicating vertically coherent wave propagation. Above approximately 30 hPa, the upward-

propagating wave activity begins to shift equatorward with height, and this meridional displacement becomes more pronounced at higher altitudes. Such latitudinal shifting of planetary wave propagation is likely modulated by the background waveguide structure (**Fig. 6a**) of the SH winter atmosphere **(Butchart et al., 1982)**. The upward- and equatorward-propagating planetary waves in the stratosphere ultimately lead to a pronounced region of E-P flux divergence in the

mid- to low-latitudes near the stratopause (1 hPa, 15°-55°S). This divergence indicates wave breaking and the deposition of westward momentum into the background flow, producing a peak deceleration anomaly of approximately $-3$ m s$^{-1}$ day$^{-1}$ - an enhancement of about 55% relative to the climatological mean ($-5.6$ m s$^{-1}$ day$^{-1}$). This anomalous momentum forcing provides a dynamical explanation for the strong negative zonal wind center ($\sim -18$ m s$^{-1}$) observed in the

midlatitude upper stratosphere in **Fig. 3**.

      However, besides the negative zonal-mean zonal wind anomalies over the midlatitudes (30°–50°S) shown in **Fig. 3**, a region of positive anomalies is also present over high latitudes (60°–75°S) near 3 hPa. This high-latitude response may be linked to the dynamical adjustment associated with the midlatitude wind anomalies. Under the quasi-geostrophic and hydrostatic balance approximation, the zonal wind can be approximated by the thermal wind **(Andrews et**

**al., 1987)**, with both the climatological mean temperature (T) and zonal wind (U) satisfying the thermal wind equation. Accordingly, the zonal thermal wind anomaly ($u_T^{'}$) and the meridional gradient of the temperature anomaly ($T^{'}$) satisfy:

$$\frac{\partial u_T^{'}}{\partial z} = -\frac{R}{aHf}\frac{\partial T^{'}}{\partial \varphi} \qquad\qquad (12)$$

where $f = 2\,\omega\,sin\,\varphi$ is the Coriolis parameter, $R$ is the gas constant for dry air, a is the Earth's radius, and H is the scale height of the atmosphere. This relationship indicates that the meridional gradient of the temperature anomaly ($\partial T^{'}/\partial\varphi$) thermodynamically forces the vertical shear of the zonal wind.

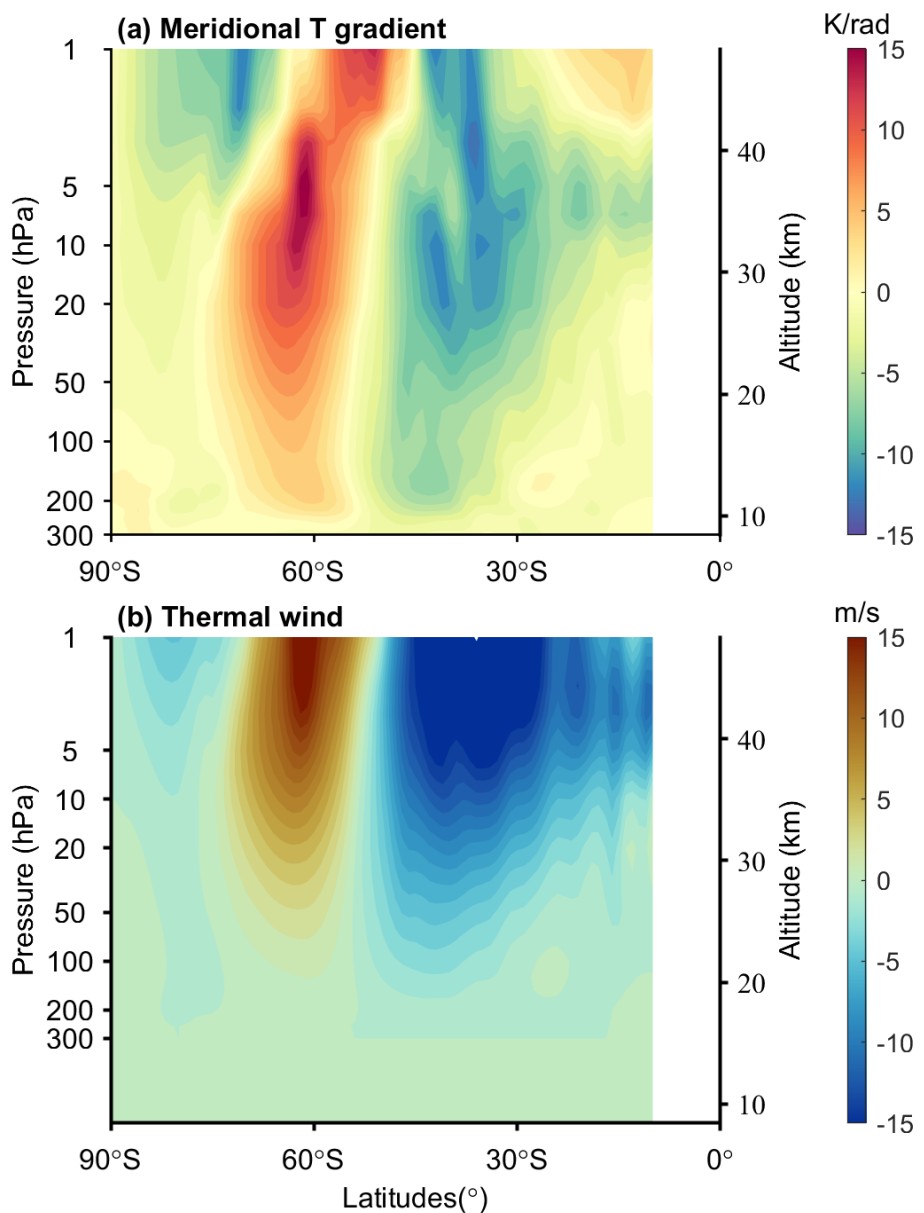

**Figure 8:** (a) Meridional gradient of temperature anomalies ($\partial T'/\partial \varphi$) during the Southern Hemisphere winter associated with positive MIOD events. (b) Corresponding zonal wind anomalies ($\boldsymbol{u'_T}$) derived from the thermal wind equation.

The positive temperature anomaly in the midlatitude stratosphere, centered near 50° S and ~10 hPa (**Fig. 3b**), induces a marked meridional asymmetry in the temperature gradient ($\partial T'/\partial \varphi$) (**Fig. 8a**), with a strong negative gradient band (−11 K rad⁻¹) between 30° S and 50° S and a positive gradient over 50° S–75° S. According to the thermal wind relationship, this gradient structure drives a typical baroclinic response in the zonal thermal wind anomalies $u'_T$ (**Fig. 8b**): the low-latitude negative gradient region forces a −20 m s⁻¹ westerly anomaly centered near the stratopause and extending downward to 100 hPa, while the midlatitude positive gradient

anomaly generates an eastward wind anomaly of up to $+12$ m s$^{-1}$ at the stratopause, and the high-latitude negative gradient corresponds to a secondary $-3.4$ m s$^{-1}$ anomaly. This thermal wind framework explains the dipolar structure of the zonal wind anomalies in **Fig. 3**, including the high-latitude reversal to positive anomalies. The smaller amplitude of the observed anomalies compared to the calculated thermal wind (e.g., $+8$ vs. $+12$ m s$^{-1}$ in mid-to-high latitudes) is

partially attributed to planetary wave breaking, as indicated by the EP flux divergence ($\sim -2$ m s$^{-1}$ d$^{-1}$ at 60° S, 1 hPa), which acts to weaken the thermal-wind-driven response.

        **Fig. 9** provides an objective view of how positive MIOD events modify the Southern Hemisphere polar vortex by examining potential vorticity (PV) anomalies on the 850-K isentropic surface. During positive MIOD events, a zonally asymmetric PV anomaly pattern

appears, with reduced PV over the high-latitude western sector, while enhanced PV appears between 30°–60°S in the eastern sector. This anomaly distribution is associated with a westward displacement of the composite vortex boundary (pink solid contour) relative to its climatological position (dashed gray circle). Such a deformation of the vortex edge represents a geometric manifestation of a stationary zonal wavenumber-1 (WN-1) perturbation, consistent with the WN-

1 geopotential height anomalies identified in **Figs. 4–5** and the associated refractive-index conditions that favor vertical propagation. The PV-based metric therefore provides a structural complement to the earlier diagnostics, illustrating how the MIOD-related wave perturbations project onto the vortex geometry.

        To sum, the most pronounced responses are linked to positive MIOD events, whereas

negative phases produce weaker atmospheric signals. This asymmetry is consistent with the diagnosed differences in planetary wave activity, with positive events favoring strong wavenumber-1 anomalies that can effectively propagate into the stratosphere, while negative events primarily excite higher-wavenumber anomalies that, according to the RI diagnostics, are not permitted to propagate vertically and therefore do not reach the stratosphere.


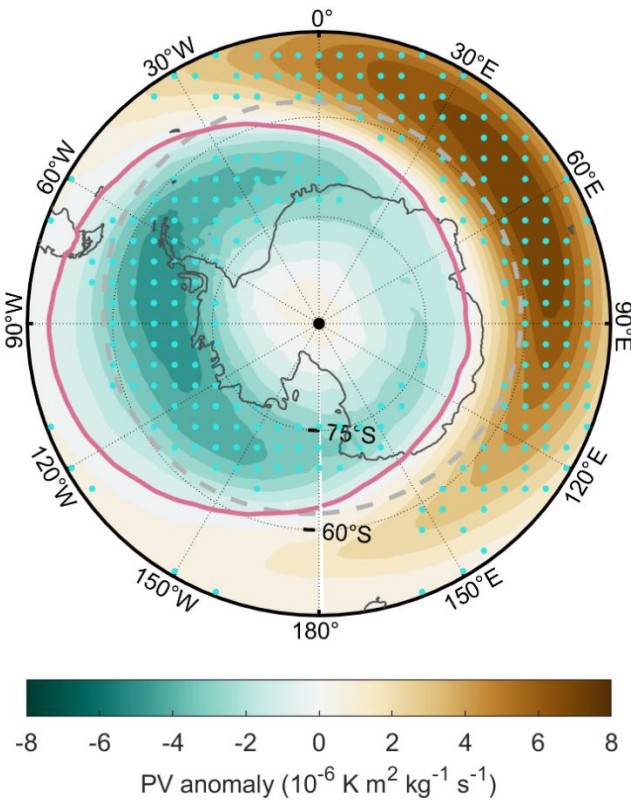

**(a) MIOD Positive Composite (JJA)**

**Figure 9:** Composite anomalies of the potential vorticity (PV) field for positive MIOD events at the 850 K isentropic level during June–August (JJA). Shading indicates the PV anomalies (units: $10^{-6}$ K m$^2$ kg$^{-1}$ s$^{-1}$), with stippling showing regions exceeding the 95% Monte Carlo significance level. The dashed gray contour marks the climatological polar vortex boundary derived from the JJA-mean PV field. The solid pink contour shows the composite vortex boundary.

Driven by the zonal momentum forcing provided by the divergence of the E-P flux in the low- and mid-latitude stratosphere, the Brewer–Dobson (B-D) circulation is intensified during positive MIOD events. **Fig. 10a** presents the residual mean meridional and vertical winds ($v^*$, $w^*$) calculated from the transformed Eulerian mean (TEM) equations (streamlines), along with percentage anomalies of ozone mixing ratio (color shading), with regions failing the 90% confidence test masked in gray. The results reveal enhanced poleward flow in the stratosphere above 10 hPa between the equator and 40° S, accompanied by strengthened downward motion between 30° and 60° S. In the tropical middle stratosphere (30-40 km), the anomalous ascending branch transports ozone produced by lower-level photochemical reactions upward toward the stratopause, generating a pronounced ozone enhancement (+4 %). Near the stratopause (~3 hPa),

the intensified poleward transport in the anomalous circulation produces a marked divergence region in the midlatitudes, leading to ozone decrease of up to –6 %. With increasing latitude, the meridional flow weakens and anomalous subsidence strengthens, forming a convergence center

in the mid-stratosphere over the midlatitudes (30 hPa, 60° S), where ozone is accumulated (+6 %), which could contribute to the local heating through shortwave absorption.

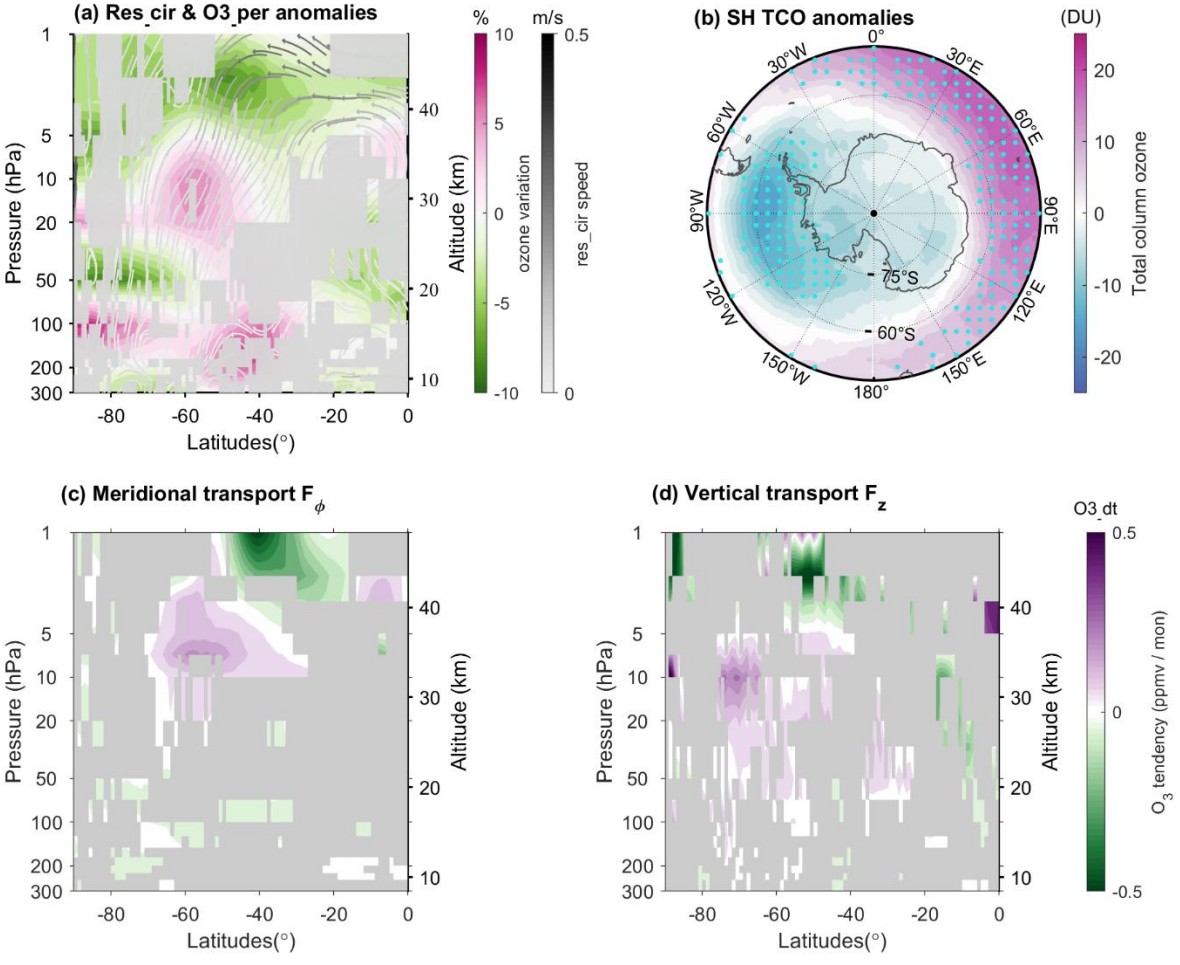

**Figure 10:** (a) Composite ozone mixing ratio percentage anomalies and residual circulation anomalies ($\bar{v}*$, $\bar{w}*$) during positive MIOD events (JJA). Gray shading denotes regions where residual circulation anomalies are not significant at the 90% level. (b)
Composite total column ozone anomalies for positive MIOD events (JJA), with stippling indicating significance at the 90% level. (c) Composite anomalies of the meridional transport term ($F\phi$). Gray shading marks regions not significant at the 90% level. (d) Same as (c), but for the vertical transport term ($Fz$).

This "tropical ascent–midlatitude descent" anomaly pattern redistributes ozone in the
latitude–height plane, with the associated changes resulting primarily from transport. It leads to

reduced ozone in the tropical lower stratosphere and the midlatitude upper stratosphere, while enhancing ozone in the midlatitude middle–lower stratosphere. As ozone concentrations typically peak near ~20 hPa, changes above 10 hPa-such as the decrease near 40°S and the concurrent tropical increase-indicate substantial anomalies in ozone transport. The increased ozone in the midlatitude stratosphere is likely an important factor contributing to the significant warming over 45°-60°S through radiative heating. However, given the potential influence of Brewer-Dobson circulation anomalies, especially in the lower stratosphere, the exact relative contributions of radiative and dynamical processes cannot be determined from the present analysis. Such ozone transport anomalies are also clearly reflected in the total column ozone (TCO) changes shown in **Fig. 10b**, with a pronounced increase of approximately 20 DU over the midlatitudes (45°–60° S) and a marked decrease at higher latitudes, particularly within the 60° W-120° E sector. These patterns are consistent with the convergence and divergence features of the anomalous circulation evident in **Fig. 10a**. The close resemblance between the TCO pattern in **Fig. 10b** and the PV anomalies on the 850-K surface (**Fig. 9**) further supports the interpretation that both fields reflect the large-scale circulation anomalies associated with the MIOD-related WN-1 perturbation.

The diagnostic transport terms further substantiate that these ozone anomalies arise primarily from dynamical redistribution rather than in situ chemistry. To quantify the contribution of large-scale dynamic transport to the ozone response, we diagnose an anomaly-based TEM transport proxy defined as:

$$T_{dyn} = -v^{*\prime}\frac{\partial[O3]}{\partial y} - \Delta w^{*\prime}\frac{\partial[O3]}{\partial z} \qquad (13)$$

where $v^{*\prime}$ and $w^{*\prime}$ are anomalies of the meridional and vertical residual velocities relative to their climatological means, and the ozone gradients are computed from the climatological zonal-mean ozone field. This diagnostic represents the anomalous dynamical transport associated with circulation anomalies and is used to construct the horizontal and vertical transport components. **Fig. 10c and 10d** show the composite anomalies of these meridional and vertical transport terms during positive MIOD events.

The meridional transport component (**Fig. 10c**) exhibits a dipole-like anomaly pattern, with negative values near the subtropical stratopause (~30° S, ~3 hPa) and positive values over the midlatitudes around ~60° S and ~10 hPa. This dipole structure indicates a strengthened poleward transport branch between these regions: the negative anomalies near 30° S, 3 hPa are consistent with tendencies that remove ozone from the subtropical stratopause, whereas the positive

anomalies near 60° S, 10 hPa reflect tendencies that add ozone into the midlatitude stratosphere, in line with the corresponding ozone anomalies. The vertical transport term (**Fig. 10d**) exhibits anomalies that are consistent with the MIOD-related residual circulation. Negative anomalies near ~50° S and ~3 hPa are consistent with an enhanced downward branch of the anomalous residual circulation (**Fig. 10a**), which tends to export of ozone-rich air from the stratopause region. At higher latitudes, the positive anomalies between ~70° S and 10-20 hPa likely reflect the corresponding downward transport of ozone into lower levels. The combined behavior of the meridional and vertical transport terms closely matches the spatial pattern of ozone tendencies, indicating an MIOD-related redistribution of ozone from the subtropical upper stratosphere toward the midlatitude lower–middle stratosphere. This dynamical interpretation accounts for the dominant features of the ozone response, although contributions from chemical processes or other factors cannot be ruled out.

## 4 Discussion: Mesospheric Extension of the MIOD Influence

The stratospheric responses described above suggest that MIOD-related perturbations may extend upward into the mesosphere, raising the question of how far the influence of MIOD projects vertically. To investigate the full vertical structure of the atmospheric response, we complement the stratospheric analysis with merged HALOE–SABER temperature observations spanning 10–100 km and SD-WACCM6 simulations. Because the free-running nature of SD-WACCM6 above ~50–60 km allows the mesosphere–lower thermosphere (MLT) variability to evolve independently of the imposed stratospheric state, the comparison between observations and model output provides a basis for examining whether the mesospheric anomalies inferred from observations are dynamically consistent with those that arise internally in the model. This framework enables us to assess potential pathways through which MIOD-related stratospheric perturbations may influence the mesosphere, without presupposing the underlying dynamical mechanism.

We use merged HALOE–SABER temperature observations (10-100 km) for 1991-2022, restricted to 55° S-equator owing to data gaps poleward of 50° S in HALOE and limited coverage south of 55° S in SABER. **Fig. 11a** shows composite zonal-mean temperature anomalies for positive MIOD events from merged HALOE-SABER observations (1991-2022), and **Fig. 11b** shows the corresponding composite constructed from SD-WACCM6 outputs.

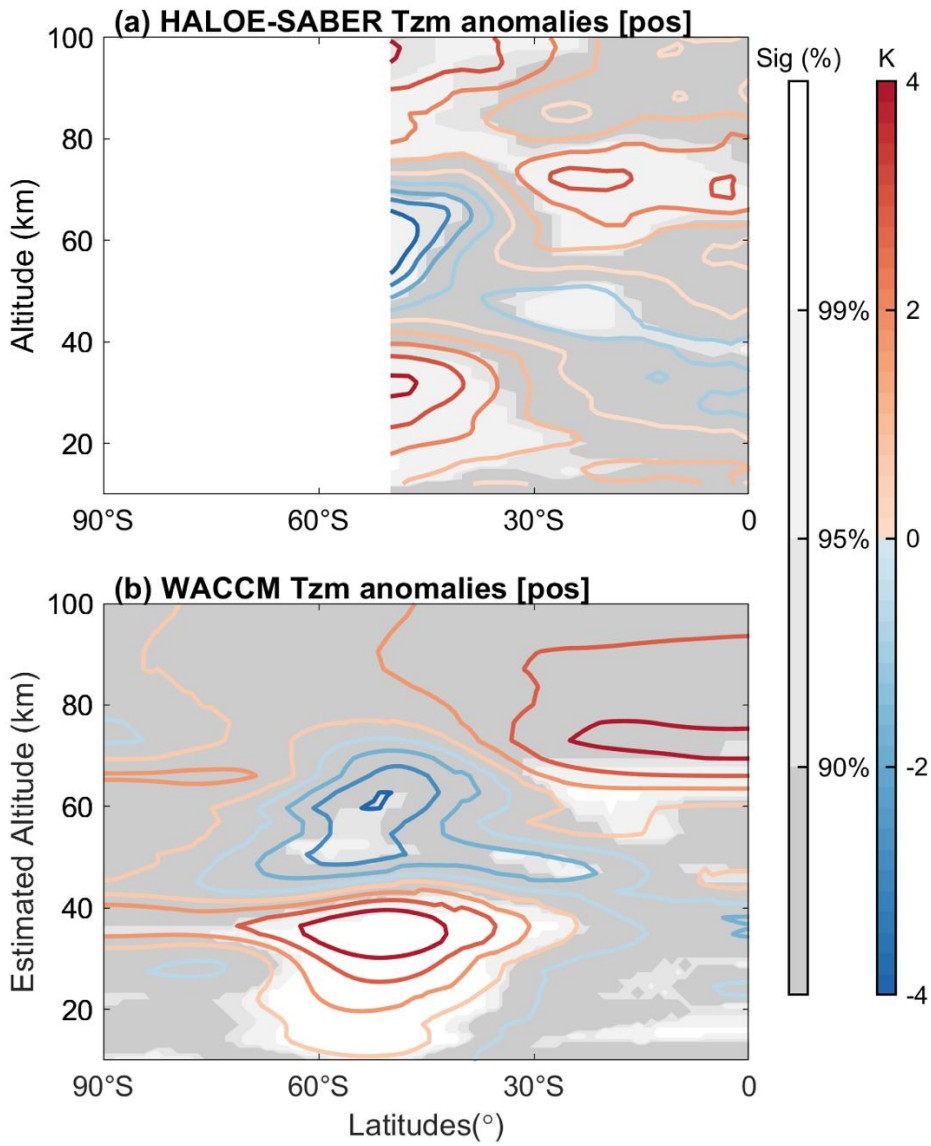

**Figure 11:** (a) Composite zonal-mean temperature anomalies for positive MIOD events from merged HALOE–SABER satellite observations (1991–2022) over 10–100 km. White, light gray, gray, and dark gray shading denote regions exceeding the 99 %, 95 %, and 90 % confidence levels, and those failing the significance test, respectively, based on a Monte Carlo method. (b) Same as (a), but from SD-WACCM6 simulations.

In the midlatitude stratosphere, satellite observations show zonal-mean temperature anomalies of +3 K at 45° S and 30-40 km, slightly weaker than the +4 K seen in the ERA5 reanalysis. This difference may be due to the satellite data assimilation period (post-1992) not covering earlier strong MIOD events. From the stratopause upward to the mesosphere (40-70 km), a significant cold anomaly develops, with a -4 K minimum centered at 50° S and a tilted

cold band of about -1 K extending from the equatorial stratopause (40 km, 0°) toward the lower mesosphere at 30° S (45-50 km). Above this level, from the upper mesosphere to the lower

thermosphere (80-100 km), the anomalies shift to warming, reaching up to +2 K. As SD-WACCM6 is constrained by reanalysis only in the stratosphere and evolves freely in the mesosphere–lower thermosphere, it provides a means to assess whether the mesospheric anomalies inferred from observations exhibit structural similarities to those generated internally. The SD-WACCM6 composites show a broadly similar large-scale structure but with weaker

amplitudes. However, the midlatitude warming in mesosphere/lower thermosphere region seen in observations is largely absent in the model, and the tropical anomaly remains below 1 K and is not statistically significant. This discrepancy between the observations and SD-WACCM6 may indicate that the processes giving rise to the upper-mesospheric and lower-thermospheric response are not fully captured, as SD-WACCM6 is not constrained in the mesosphere. An

additional contributing factor may be the non-overlapping portions of the observational record (1991–2022) and the model simulation period used here. Despite observational gaps at high southern latitudes, both the satellite observations and SD-WACCM6 simulations exhibit broadly similar large-scale vertical anomaly patterns extending form the stratosphere into the mesosphere.

Given that stratospheric disturbances may project upward through dynamical pathways, it is instructive to consider how the zonal wind anomalies associated with MIOD events modify the background flow. During positive MIOD events, the stratospheric zonal wind anomalies exhibit a distinct dipole pattern, characterized by a pronounced weakening of the midlatitude westerlies ($\Delta u \approx -18$ m s$^{-1}$). Such a wind structure is expected to strongly modulate gravity-wave critical-

level filtering, because changes in the strength of the background westerlies determine whether and where waves with different horizontal phase speeds satisfy the critical-level condition and are preferentially absorbed in the lower atmosphere. Since SD-WACCM6 is not constrained in the mesosphere, the simulated mesospheric temperature response reflects internally generated variability that resembles the observed anomalies, allowing the model's dynamical process to be

used diagnostically to explore how MIOD-related stratospheric perturbations may project upward into the mesosphere.

In the positive MIOD composite, the SD-WACCM6 gravity wave drag (GWD) anomalies show a significant eastward acceleration of up to +12 m s$^{-1}$ d$^{-1}$ near 0.1hPa (60km) at midlatitudes, accompanied by a westerly wind anomaly (+8 m s$^{-1}$) at 0.01 hPa around 30° S (**Fig.**

**12**). These features are consistent with a scenario in which a larger fraction of westward-propagating gravity waves is absorbed by the stratospheric zonal wind, so that the remaining upward-propagating spectrum deposits a net eastward momentum flux in the upper mesosphere. At high southern latitudes (60°–80° S), no clear gravity wave drag anomalies are found, and the zonal wind perturbations above 60 km remain weak. These results indicate that the MIOD-

related stratospheric wind anomalies imposed through nudging are sufficient to modify gravity-wave filtering in the free-running mesosphere of SD-WACCM6, leading to temperature and wind responses that broadly resemble those inferred from observations. This suggests that the upward influence of MIOD events likely operates through this dynamical pathway, with the midlatitudes providing the most efficient conduit for transmitting stratospheric perturbations into

the mesosphere.

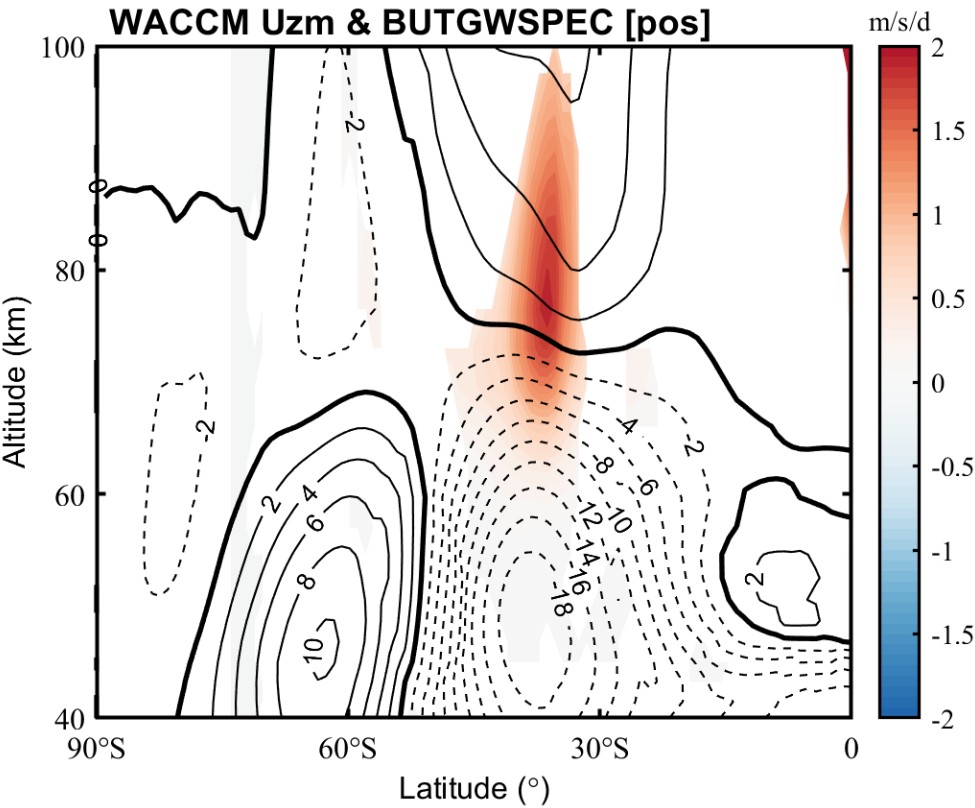

**Figure 12:** Composite anomalies of JJA zonal mean zonal wind and gravity wave drag in the Southern hemispheric upper stratosphere to the lower thermosphere during positive MIOD events. Contours denote composite anomalies of zonal wind (solid for positive/eastward anomalies and dashed for negative/westward anomalies), with a contour interval of 2 m s⁻¹. Shading

indicates anomalies of gravity wave drag, and only regions passing the 90% Monte Carlo confidence test are shown.

The diagnosed stratospheric response to MIOD events exhibits a clear phase asymmetry that can be traced primarily to differences in the dominant planetary-wave forcing. Positive MIOD events preferentially excite zonal wavenumber-1 disturbances that are able to propagate upward under austral winter background conditions, whereas negative events are dominated by higher-wavenumber anomalies that remain largely confined below the tropopause. The associated EP-flux divergence provides momentum forcing that strengthens the residual circulation, contributing to midlatitude ozone redistribution through enhanced poleward transport and downwelling. Deviations between thermal-wind estimates and reanalysis winds further point to a dynamical contribution from planetary-wave forcing, although diabatic, radiative, and chemical processes may also play a role.

In the mesosphere, SD-WACCM6 produces a response that is structurally similar to satellite observations. Within the model, MIOD-related stratospheric wind anomalies modulate gravity-wave filtering and wave-mean flow interactions, leading to coherent mesospheric drag and circulation anomalies. While discrepancies persist, particularly at higher altitudes, these results indicate that gravity-wave filtering provides a physically plausible pathway linking MIOD-related stratospheric disturbances to the mesospheric response.

**5 Conclusion**

The present study highlights the significant role of SH winter Indian Ocean SST anomalies in modulating the circulation and thermal structure of the middle and upper atmosphere. By constructing a new index based on the leading EOF mode of JJA SST variability, we show that positive MIOD events are accompanied by pronounced planetary-wave responses, enhanced upward EP flux, and a strengthening of the residual meridional circulation. These dynamical anomalies warm the midlatitude stratosphere, modify the vertical and meridional structure of the polar vortex, and redistribute ozone toward the midlatitudes.

The atmospheric response exhibits a clear phase asymmetry: positive MIOD events favor strong zonal wavenumber-1 anomalies that are able to propagate into the stratosphere under austral winter conditions, whereas negative phases primarily excite higher-wavenumber disturbances that, according to the refractive-index diagnostics, remain confined below the tropopause and produce a much weaker stratospheric signal. As a consequence, the circulation and thermal anomalies associated with positive MIOD events project onto a SAM-negative-like

pattern, characterized by polar-vortex weakening and a broader hemispheric circulation adjustment.

Satellite observations and SD-WACCM6 simulations further indicate that MIOD-related anomalies extend into the Southern Hemisphere mesosphere, with the model suggesting a role for gravity-wave drag modulation in linking stratospheric wind anomalies to the mesospheric response. The MIOD-related atmospheric signal identified here indicates that Indian Ocean SST variability acts as an additional source of large-scale dynamical variability in the Southern Hemisphere, complementing established influences such as ENSO and the QBO, and highlighting a previously underappreciated pathway through which tropical ocean variability affects the middle and upper atmosphere on interannual timescales.

## Acknowledgement

This work was supported by the National Natural Science Foundation of China Grants 42130203, 42241115, 42275133, and 42241135, and the National Key R\&D Program of China Grant 2022YFF0503703

## Data availability

HALOE satellite observations (1991–2004) are available from the NASA Goddard Earth Sciences Data and Information Services Center (GES DISC) at https://disc.gsfc.nasa.gov/. SABER temperature data (2002–2020) are provided by GATS Inc. and can be accessed via https://saber.gats-inc.com/data.ph. ERA5 reanalysis data are available from the Copernicus Climate Data Store at https://cds.climate.copernicus.eu/. WACCM6 simulations are distributed by the CESM project and can be obtained through the Earth System Grid Federation at https://esgf-node.llnl.gov/projects/esgf-cesm/. The midlatitude Indian Ocean Dipole (MIOD) index constructed in this study is derived from the Hadley Centre Sea Ice and Sea Surface Temperature dataset (HadISST; https://www.metoffice.gov.uk/hadobs/hadisst/). TOMS ozone data are publicly available at https://ozonewatch.gsfc.nasa.gov/meteorology/SH.html. Solar F10.7 flux data are available from NASA's OMNIWeb archive at https://omniweb.gsfc.nasa.gov/. The derived data used for generating the figures displayed in this article are available on (Yang, 2025).

## Author contribution

CY, XG, and TL planned the campaign; CY, XG, XW, JZ, and XF performed the measurements; XY, XG, XW, and XF analyzed the data; CY wrote the manuscript draft; TL, XW, JZ, XF, and XX reviewed and edited the manuscript.

## Competing interests

The authors declare that they have no conflict of interest.

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
