# Peer review of "Impact of the Indian Ocean Sea Surface Temperature on the Southern Hemisphere Middle Atmosphere"

_EGUsphere, 2025_

## Author Comment (AC1)

*This manuscript introduces a new climate index, the "Middle-latitude Indian Ocean Dipole" (MIOD), and investigates its influence on the Southern Hemisphere's middle and upper atmosphere. The study uses a multi-faceted approach combining reanalysis, satellite data, and model simulations to build a compelling narrative. The proposed physical mechanism links positive MIOD events to enhanced planetary wave activity, which in turn drives significant stratospheric and mesospheric changes. This mechanism is logical, well-articulated, and represents a potentially significant contribution to our understanding of ocean-atmosphere coupling. The finding of a strong asymmetry in the atmospheric response between positive and negative MIOD events is particularly noteworthy.*

*However, the manuscript in its current form is undermined by several methodological flaws and a lack of careful preparation that question the validity and reproducibility of its core findings. Most central is the pre-removal of EESC from SST prior to the EOF, which is unusual and risks biasing the MIOD pattern. Second is event selection that is vulnerable to ENSO aliasing (e.g., 2016). Furthermore, the statistical robustness is limited by a small sample size, and errors in figure labels and captions detract from the paper's credibility. With stronger methodological circumspection, a set of focused robustness checks, and cleaner presentation, the paper can reach the level the idea deserves.*

*All my concerns are detailed below. I do not necessarily expect the authors to address every point, but I do expect the critical issues to be dealt with convincingly for the work to be credible.*

**Major comments**

*1) SST preprocessing with EESC before the EOF*

*The manuscript removes EESC from JJA SST prior to the EOF but largely treats this as routine. It is not. EESC is a stratospheric halogen proxy : a direct, widely accepted causal pathway to basin-scale SST variability is not established. Regressing out a non-linear, parabolic-like trajectory from SST can reshape low-frequency variance and therefore the EOF structures themselves. In other words, the MIOD pattern may be sensitive to this step. If the intention is to isolate an SST pattern "untainted" by ozone-related radiative trends, that needs a clear physical rationale. Otherwise, a standard approach is to detrend SST (and, if desired, apply ENSO/SIOD partialing in atmospheric fields, not in SST itself). At minimum the preprocessing must be made prominent in the figure caption and methods, and the results shown to be robust to its omission.*

**Response:** We thank the reviewer for this constructive comment. We agree that regressing SST onto EESC is not a standard procedure and may unintentionally reshape low-frequency SST variance. Our original intention was simply to isolate the interannual SST variability associated with the MIOD pattern, but we recognize that using a stratospheric halogen proxy is not physically justified for SST preprocessing.

To avoid introducing any unnecessary assumptions, we have **completely**

**removed the EESC preprocessing step**. In the revised manuscript, the EOF analysis is performed directly on JJA SST anomalies relative to the 1980–2020 climatology, which represents the standard approach for examining interannual SST variability in climate studies.

We have verified that removing the EESC preprocessing does not materially affect the results:

a. The revised EOF2 spatial pattern (the MIOD mode) has a pattern correlation of 0.8 with the original version.

b. The identified MIOD event years differ by only one year on each side.

c. The composite atmospheric circulation and temperature responses remain nearly unchanged.

[Figure]

**Figure R1.** SST patterns of (a) EOF1 and (b) EOF2, and principal component time series of (c)
EOF1 and (d) EOF2, derived from Indian Ocean SST anomalies during austral winter (JJA) for
1980–2020 over the domain 60°S–5°N, 40°E–145°E.

The updated EOF patterns and PCs are shown in Figure 1 of the revised manuscript. In the updated analysis, EOF1 and EOF2 explain 21.3% and 13.4% of the total variance, respectively. These tests demonstrate that the MIOD pattern and its associated atmospheric impacts are robust to the choice of SST preprocessing. Notably, removing the EESC step has only a minor effect on the identification of MIOD years—the revised positive-event list differs by only one year from the original selection. The details of this comparison are provided in our response to Comment 2.

**Corresponding Revisions in the manuscript:**

- The previous description of detrending and EESC regression (former Eq. 9 and related text in Section 2.1) has been removed.
- The Methods section now states that the EOF analysis is applied directly to **JJA SST anomalies without detrending or EESC adjustment**.
- The figure caption of Fig. 1 has been updated to reflect the revised preprocessing.
- EESC is now discussed only in the context of long-term changes in the atmospheric circulation and temperature fields, and is no longer used in the SST processing or in the definition of the MIOD index.

*2) Event selection and ENSO aliasing*

*The paper aims to separate MIOD impacts from ENSO, but the threshold-based exclusion (JJA Niño-3.4 ±1σ) is a blunt tool. A case in point is 2016: the trailing influence of the 2015–16 El Niño plausibly persists into mid-2016, yet 2016 enters the "positive MIOD" set. Given the small sample, one influential year can strongly color the composites in Fig. 3. Threshold exclusion is weaker than regression-based control. The latter is standard and makes better use of the record. At a minimum the reader needs to see a 2016-excluded positive composite and a regression-controlled view to judge robustness.*

**Response:** Thanks for this helpful comment. You noted that excluding ENSO years solely by applying a JJA Niño-3.4 ±1σ threshold may not fully separate MIOD from ENSO influences, especially for 2016, which may still carry residual effects from the strong 2015-2016 El Niño.

After revising our SST preprocessing to use JJA SST anomalies directly (see

Response to Comment 1), the temporal evolution of the MIOD index also changed slightly. As a result, **2016 is no longer selected as a positive MIOD year**, even under the original ±1σ threshold method. The revised MIOD event lists differ from the original submission by only one event on each side.

To further assess robustness, we followed the reviewer's suggestion and performed an additional test by **removing the linear influence of ENSO from the**

**MIOD index**. Specifically, we regressed the MIOD index onto the JJA Niño-3.4

index and repeated the event identification using the ENSO-removed residual. This regression-based selection yields a set of MIOD years that is nearly identical to the threshold-based set, again differing by only one event (**Figure 2R**).

[Figure]

**Figure R2:** (a) Austral winter (JJA) MIOD index with the linear Niño-3.4 contribution removed
(black) and the corresponding PC2 time series, with positive (negative) PC2 values shaded in red
(blue). (b) Two-line timeline summarizing MIOD event selection. Gray bars mark ENSO years.
Red and blue bars represent positive and negative MIOD anomalies, respectively, while gray bars
in the MIOD row indicate years in which MIOD events overlap with ENSO and are thus excluded
from the independent MIOD composites.

Importantly, the composite atmospheric circulation and temperature responses remain essentially unchanged across all sensitivity tests of the revised threshold-based selection and the regression-controlled MIOD index.

These results indicate that the MIOD-related atmospheric signals are not sensitive to whether 2016-or any ENSO-influenced year-is included, and that MIOD and ENSO impacts are effectively separated for the purposes of this study.

We have updated Figure 2 (event selection) and the related text in the manuscript to reflect the revised MIOD years as "As a robustness check, we also removed the linear influence of ENSO by regressing the MIOD index onto the JJA Niño-3.4 index before identifying events. The resulting MIOD years were nearly identical to those obtained using the threshold-based approach, differing by only one positive event".

The MIOD years used in the final analysis are:

**Positive MIOD:** 1984, 1992, 1996, 2002, 2005

**Negative MIOD:** 1981, 1986, 1993, 2006, 2017, 2018, 2019

It is true that both the positive and negative MIOD are modest in number, which limits statistical power. Nevertheless, all reasonable perturbations to events selection (leave-one-out removal, Niño-3.4 regression control, and threshold perturbations) converge on the same qualitative asymmetry, indicating that no single year is disproportionately influencing the results.

*3) Positive–negative asymmetry: mechanism and power*

*The descriptive evidence for asymmetry is good (Fig. 5), but the paper stops short of explaining why the SST patterns in Fig. 4 project so differently onto the large-scale wave field. There is room, and need, for a more mechanistic line: stationary-wave sources/diabatic heating anomalies, Charney–Drazin refractive index/waveguide diagnostics, or MIOD→WN-1 amplitude regressions would move the argument beyond "constructive vs destructive interference". The negative-event null should also be tempered by an explicit acknowledgement of limited power (7 cases) and supported by leave-one-out and threshold-sensitivity checks. A brief discussion of MIOD's relationship to the SAM would give useful context for vertical propagation and annular-mode fingerprints.*

**Response:** We appreciate your suggestion to further substantiate the mechanism underlying the positive-negative asymmetry. To determine whether the planetarywave anomalies in the composites originate from the Indian Ocean SST forcing, we include the Takaya–Nakamura (TN) wave-activity flux to diagnose the stationarywave response and its source region.

[Figure]

**Figure R3 (Figure 4 in the revised manuscript):** Composite anomalies for positive MIOD

events during June–August (JJA). (a) Sea surface temperature (SST) anomalies (shading), with only regions passing the 90% Monte Carlo confidence test shown. Overlaid contours indicate zonal anomalous geopotential height at 850 hPa, with orange (blue) lines representing positive (negative) anomalies. Contours are bolded where the anomalies are statistically significant at the

90% confidence level. Black arrows show the Takaya-Nakamura wave activity flux (TN flux) at hPa, illustrating the horizontal propagation of anomalous planetary wave activity. (b) is the same as (a) but for composite of negative events.

As shown in Figure 3R (Fig. 4 in the revised manuscript), the MIOD SST dipole modifies lower-tropospheric thermal contrast and diabatic heating, providing a planetary wave source over the midlatitude Indian Ocean. Consistent with this forcing, both positive and negative MIOD events generate clear TN-flux anomalies at hPa (**Figure 3R**), indicating that the SST pattern projects onto the large-scale wave field in both phases. However, the associated geopotential height anomalies reveal a fundamental structural difference: positive MIOD events are dominated by a zonal wavenumber-1 pattern, whereas negative MIOD events project mainly onto wavenumber-3.

This distinction is crucial for vertical propagation. As shown by the Charney–Drazin refractive index diagnostics (**Figure 4R**), the JJA waveguide from the upper troposphere to the lower stratosphere favors the propagation of WN-1, while WN-3 is strongly refracted or trapped below the tropopause. Thus, the null result for negative events should not be interpreted as an absence of wave forcing, but rather as ineffective vertical transmission due to the unfavorable WN-3 structure.

The related description and discussion are added in lines 438-448 of the revised manuscript as "*To access whether the Indian SST anomalies and the global geopotential anomalies are associated, we further examined the TN wave-activity flux at 850 hPa (as indicated by the vector in Fig. 4). For either positive or negative MIOD cases, TN-flux perturbations extend from the midlatitude eastward to the south Pacific, indicating that SST anomalies in either events can modulate the large-scale wave field. The similarity of the TN-flux patterns between positive and negative MIOD events indicates that both phases are capable of exciting planetary-wave activity over the midlatitude Indian Ocean. Thus, the contrasting atmospheric responses between the two phases are unlikely to arise from differences in the strength or spatial extent of the planetary -wave forcing itself. Instead, this result suggests that the differences in the structure and phase of the associated geopotential height anomalies may motivates a more targeted diagnosis of the planetary-wave characteristics*."

Finally, as suggested by the reviewer, we added a brief discussion in the revised manuscript noting that the enhanced upward propagation of WN-1 during positive MIOD events weakens the stratospheric polar vortex and therefore projects onto the negative phase of the Southern Annular Mode (SAM) as "*The zonal wind and*

*temperature anomalies (weakened midlatitude westerlies, strengthened high-latitude*

*westerlies, and polar-cap warming) closely resemble the canonical negative phase of*

*the Southern Annular Mode (SAM)*" in lines 379-381 of the revised manuscript. This

SAM-like pattern is included in the revised manuscript as a familiar dynamic fingerprint of the diagnosed stratospheric circulation anomalies.

[Figure]

**Figure R4 (also as Figure 6 in revised manuscript)**: June–July–August (JJA) climatological
refractive index (RI) for (a) zonal wavenumber-1 (WN-1) and (b) zonal wavenumber-3 (WN-3),
averaged over 1979–2020. Negative RI values are masked and shown in gray. The pressure axis is
plotted on a logarithmic scale.

*4) SD-WACCM6 framing*

*The SD configuration is nudged to reanalysis: it provides diagnostic consistency (e.g.,*
*gravity-wave drag, MLT structure) rather than an independent forced response. The*
*manuscript sometimes reads as if the model "confirms" the mechanism. It would be*
*more accurate to present SD-WACCM6 as a way to diagnose fields not available in*
*reanalysis, with language calibrated accordingly. If any free-running sensitivities or*
*prior literature exist that align with the sign/structure of the MLT anomalies, pointing*
*to them would help.*

**Response:** Thank you for this important clarification. We fully agree that, because the

SD-WACCM6 configuration is nudged toward reanalysis in the troposphere and stratosphere, it should not be interpreted as an independent simulation of an MIOD- forced response. We have revised the section 4 in the manuscript accordingly to avoid any language suggesting model "confirmation" of the mechanism.

To clarify, nudging in SD-WACCM6 is confined to the lower and middle atmosphere (approximately below 50–60 km), and the mesosphere–lower thermosphere (MLT) remains free-running. The mesosphere response discussed in the manuscript (e.g., gravity-wave drag, residual circulation, and thermal anomalies)

therefore represent internally generated variability that is dynamically conditioned bybut not prescribed by-the imposed stratospheric anomalies associated with the MIOD.

In light of this, SD-WACCM6 is now presented solely as a diagnostic tool that provides access to dynamical quantities and vertical structures not available from reanalysis (such as gravity-wave drag or mesospheric zonal wind). The revised manuscript calibrates the language to emphasize this diagnostic role and remove any implication of causal validation.

We also note in the Discussion that the simulated mesospheric response shows structural consistency with HALOE-SABER observations, which supports the plausibility of the proposed upward-coupling mechanism without treating SD-

WACCM6 as an independent forcing experiment.

*5) Temporal evolution and breadth of robustness*

*The proposed pathway invites questions about onset/persistence and seasonality.*
*Lead–lag views (MAM→JJA→SON) would clarify timing and any spring imprint,*
*and a second reanalysis (JRA-55, MERRA-2) for key figures would demonstrate that*
*results are not a one-dataset artifact. Claims about vortex "morphology" would*
*benefit from simple, objective metrics (PV or geopotential on an isentrope; centroid,*
*ellipticity, equivalent area).*

**Response:** We appreciate the reviewer's suggestion to examine the temporal evolution and robustness of the diagnosed MIOD influence. To address this, we performed additional lead-lag composite analyses and reanalysis cross-validation.

**(1) Lead–lag seasonal evolution (MAM → JJA → SON).**

We conducted composite analyses for MAM and SON using the same MIOD

event years. In MAM, only very weak midlatitude stratospheric temperature and zonal wind anomalies were detected (<1 K and < 3 m/s), most of which are not significant (**Figure R5**). In SON, a vortex-weakening pattern (high-latitude warming and decreased zonal winds near 60°S) was apparent, but generally lacked statistical significance (**Figure R6**).

[Figure]

**Figure R5:** (a) Composite zonal-mean zonal wind anomalies for positive MIOD events during
MAM. Contours represent wind anomalies, with the dashed contour denoting the zero line.
Shading indicates statistically significant regions based on a Monte Carlo test. (b) Same as (a), but
for the zonal mean temperature anomalies. (c) and (d), similar to (a) and (b) but for negative
MIOD events.

These results indicate that the MIOD influence is most dynamically organized during JJA, consistent with the seasonality of the SH wintertime waveguide and planetary-wave transmission. For this reason, and to maintain a clear scientific narrative, we do not include them in the revised manuscript, but summarize them here as here as part of the robustness assessment here.

[Figure]

**Figure R6:** Same as Figure R5, but for composite of SON.

**(2) Cross-reanalysis robustness (ERA5 vs. MERRA-2).**

To assess dataset sensitivity, we repeated the key JJA composites using

MERRA-2. The spatial structure and amplitude of the temperature, zonal-wind, and planetary-wave anomalies closely resemble those in ERA5 (**Fig. R7**), indicating that the main results are not dependent on a single reanalysis product.

[Figure]

**Figure R7:** (a) Composite zonal-mean zonal wind anomalies for positive MIOD events based on MERRA2 datasets. Contours represent wind anomalies, with the dashed contour denoting the zero line. Shading indicates statistically significant regions based on a Monte Carlo test. (b) Same as (a), but for the zonal mean temperature anomalies. (c) and (d), similar to (a) and (b) but for negative MIOD events.

**(3) Objective vortex morphology diagnostics**

As you suggested, we evaluated PV composites on the 850-K isentropic surface. Positive MIOD events display a clear zonally asymmetric PV anomaly and a westward displacement of the vortex boundary (as shown in Fig. 9 of the revised manuscript), consistent with the stationary WN-1 response diagnosed from the geopotential height and wave-activity fields. The related discussion is added in lines 573-584 of the revised manuscript as "*Fig. 9 provides an objective view of how*

*positive MIOD events modify the Southern Hemisphere polar vortex by examining*

*potential vorticity (PV) anomalies on the 850-K isentropic surface. During positive*

*MIOD events, a zonally asymmetric PV anomaly pattern appears, with reduced PV*

*over the high-latitude western sector, while enhanced PV appears between 30°–60°S*

*in the eastern sector. This anomaly distribution is associated with a westward*

*displacement of the composite vortex boundary (pink solid contour) relative to its*

*climatological position (dashed gray circle). Such a deformation of the vortex edge*

*represents a geometric manifestation of a stationary zonal wavenumber-1 (WN-1)*

*perturbation, consistent with the WN-1 geopotential height anomalies identified in*

*Figs. 4–5 and the associated refractive-index conditions that favor vertical*

*propagation. The PV-based metric therefore provides a structural complement to the*

*earlier diagnostics, illustrating how the MIOD-related wave perturbations project*

*onto the vortex geometry."*

In contrast, negative-event PV anomalies are weak, spatially incoherent, and generally not statistically significant, and the inferred vortex boundary shows no meaningful displacement relative to the climatology. Because this does not constitute a dynamically interpretable signal, we chose to present only the positive-event PV composite in the main text.

We think these additional analyses address the reviewer's robustness concerns by demonstrating the seasonal dependence of the MIOD influence, reproducibility across reanalysis products, and objective vortex-shape diagnostics that corroborate the structure of the JJA response.

[Figure]

**(a) MIOD Positive Composite (JJA)**

PV anomaly ($10^{-6}$ K m$^2$ kg$^{-1}$ s$^{-1}$)

**Figure R8 (Figure 9 in the revised manuscript):** Composite anomalies of the potential vorticity (PV) field for positive MIOD events at the 850 K isentropic level during June–August (JJA). Shading indicates the PV anomalies (units: 10−6 K m2 kg−1 s−1), with stippling showing regions exceeding the 95% Monte Carlo significance level. The dashed gray contour marks the climatological polar vortex boundary derived from the JJA-mean PV field. The solid pink contour shows the composite vortex boundary.

**6) Ozone transport vs chemistry and gravity-wave filtering evidence**

*The TCO/ozone anomalies are interpreted primarily as transport. Where available in SD-WACCM6-SD, an ozone tendency decomposition (transport vs chemistry) or at least correlations with residual vertical velocity would strengthen that interpretation. For the MLT, the gravity-wave filtering story is plausible. If SABER gravity-wave potential energy proxies or related diagnostics can be composited, they would provide a welcome observational cross-check.*

**Response:** Thank you for the helpful suggestions to further substantiate the interpretation of MIOD-related ozone anomalies as being primarily transport driven.

(1) Stratospheric ozone: strengthening the dynamical transport interpretation

All stratospheric ozone diagnostics in the manuscript are based solely on reanalysis ozone and winds rather than SD-WACCM6 output. Building on the reviewer's suggestion, we now include an observationally constrained TEM-style transport proxy computed from anomalous residual circulation multiplied by the climatology ozone gradients:

$$T_{dyn} = -v^{*\prime}\frac{\partial [O3]}{\partial y} - \Delta w^{*\prime}\frac{\partial [O3]}{\partial z}$$

where $v^{*\prime}$ and $w^{*\prime}$ denote the anomalies of the TEM meridional and vertical residual velocities relative to their climatological means, and $\frac{\partial [O3]}{\partial y}$ and $\frac{\partial [O3]}{\partial y}$ are taken from the climatological zonal-mean ozone field. **This diagnostic therefore quantifies the anomalous dynamical transport associated with circulation anomalies, without relying on any model-derived tendency terms.**

MIOD-related transport anomalies are then obtained by compositing $T_{dyn}$ over positive MIOD years, and these composites closely resemble the corresponding TCO and lower-stratospheric ozone anomalies (new Fig. 10), reinforcing the interpretation that the observed ozone responses arise predominantly from anomalous dynamical transport rather than chemistry. Because the reanalysis does not provide full ozone tendency partitions, this TEM-based diagnostic serves as a practical and robust observational constraint. The related discussion is added in the revised manuscript as "*The diagnostic transport terms further substantiate that these ozone anomalies arise primarily from dynamical redistribution rather than in situ chemistry. To quantify the contribution of large-scale dynamic transport to the ozone response, we diagnose an anomaly-based TEM transport proxy defined as:*

$$T_{dyn} = -v^{*\prime}\frac{\partial [O3]}{\partial y} - \Delta w^{*\prime}\frac{\partial [O3]}{\partial z} \qquad (13)$$

*where $v^{*\prime}$ and $w^{*\prime}$ are anomalies of the meridional and vertical residual velocities relative to their climatological means, and the ozone gradients are computed from the climatological zonal-mean ozone field. This diagnostic represents the anomalous dynamical transport associated with circulation anomalies and is used to construct the horizontal and vertical transport components. Fig. 10c and 10d show the composite*

*anomalies of these meridional and vertical transport terms during positive MIOD*

*events.*

*The meridional transport component (Fig. 10c) exhibits a dipole-like anomaly*

*pattern, with negative values near the subtropical stratopause (~30° S, ~3 hPa) and*

*positive values over the midlatitudes around ~60° S and ~10 hPa. This dipole*

*structure indicates a strengthened poleward transport branch between these regions:*

*the negative anomalies near 30° S, 3 hPa are consistent with tendencies that remove*

*ozone from the subtropical stratopause, whereas the positive anomalies near 60° S,*

*10 hPa reflect tendencies that add ozone into the midlatitude stratosphere, in line with*

*the corresponding ozone anomalies. The vertical transport term (Fig. 10d) exhibits*

*anomalies that are consistent with the MIOD-related residual circulation. Negative*

*anomalies near ~50° S and ~3 hPa are consistent with an enhanced downward*

*branch of the anomalous residual circulation (Fig. 10a), which tends to export of*

*ozone-rich air from the stratopause region. At higher latitudes, the positive anomalies*

*between ~70° S and 10-20 hPa likely reflect the corresponding downward transport*

*of ozone into lower levels. The combined behavior of the meridional and vertical*

*transport terms closely matches the spatial pattern of ozone tendencies, indicating an*

*MIOD-related redistribution of ozone from the subtropical upper stratosphere toward*

*the midlatitude lower–middle stratosphere. This dynamical interpretation accounts*

*for the dominant features of the ozone response, although contributions from*

*chemical processes or other factors cannot be ruled out*" in lines 647-664

To avoid confusion regarding the role of SD-WACCM6, we emphasize that the model is used only to examine the possible upward influence of MIOD-induced stratospheric dynamical perturbations on the mesosphere–lower thermosphere (e.g., gravity-wave filtering and mesospheric thermal responses). Because SD-WACCM6 is nudged to reanalysis winds and temperatures in the stratosphere, its stratospheric circulation is not freely evolving. A model-based ozone tendency decomposition (transport vs. chemistry) would therefore not constitute an independent diagnosis of ozone variability and would be difficult to interpret physically. For this reason, all stratospheric ozone diagnostics and interpretations rely exclusively on reanalysis data, while SD-WACCM6 is used only for quantities that are not available from reanalysis and for exploring the upward dynamical coupling into the mesosphere.

(2) Mesosphere–lower thermosphere: observational support for gravity-wave filtering

While we agree that such diagnostics would be valuable, SABER's observational constraints limit the feasibility of constructing statistically meaningful MIOD

composites. In particular, SABER's Southern Hemisphere sampling window begins only in mid-July each year, and the SABER data record overlaps with only two robust

MIOD positive/negative event pairs (2002 and 2005). This small sample size prevents reliable isolation of MIOD-related signals from other sources of interannual variability such as ENSO, QBO, or volcanic influences. Furthermore, GWPE provides information on wave amplitude but not propagation direction, and thus cannot independently diagnose gravity-wave drag.

Nevertheless, we analyzed detrended SABER GWPE anomalies for the available years as qualitative case studies. Both 2002 and 2005 exhibit reduced GWPE above

~60 km in the winter midlatitudes during mid-July to late August, consistent with stronger filtering by enhanced stratospheric westerlies during positive MIOD

conditions. Although these examples do not allow statistical attribution, they provide observationally grounded, non-conclusive support for the plausibility of the proposed filtering mechanism. The corresponding GWPE plots are included in the Supplement for completeness and transparency, and the manuscript refers to them only as qualitative evidence.

A detailed clarification of the diagnostic role of SD-WACCM6 in the MLT is provided in our response to Comment 4, and the revised manuscript has been calibrated accordingly.

[Figure]

**Figure R9.** Mean detrended anomalies of SABER-derived gravity wave potential energy (PE) for
2002 and 2005. Anomalies are computed relative to the 2002–2022 climatology and after
removing the linear trend. Gray shading indicates missing data.

**Minor comments**

*The caption (Line 502) identifies the plot as showing TCO for negative MIOD events,*
*but the pattern shown is a direct and obvious consequence of the circulation changes*
*described for positive events in Figure 8a. The caption and text must be reconciled*
*with the figure's content.*

**Response:** Thank you. The caption was mislabeled and has been corrected.

*The x-axis of Figure 6 (both panels) is incorrectly labeled "Longitude (°)." As this is a*
*zonal-mean plot, the axis must be corrected to "Latitude (°)."*

**Response:** Thank you. The axis label has been corrected to "Latitude (°)" in the revised
figure.

*The numbering is incorrect and inconsistent in Section 2.2. There are two equations*
*labeled (4), a jump from (5) to (9), and an unnumbered thermal wind equation. Please*
*correct all numbering to be sequential.*

**Response:** We thank the reviewer for pointing this out. The equation numbering in Section 2.2 has been corrected to be fully sequential. The duplicated label (4) has been removed, the jump from (5) to (9) has been fixed, and the thermal wind equation is now properly numbered as Eq. (10).

*Figure 2b Visualization: The overlapping symbols are confusing and inefficient for conveying the event selection process. This figure should be replaced with a clearer visualization, such as a timeline or a table.*

**Response:** Figure 2b has been completely redesigned. The previous overlapping symbols have been removed and replaced with a two-line timeline visualization that clearly distinguishes ENSO years from positive and negative MIOD anomaly years. The revised timeline avoids symbol overlap, improves readability, and more effectively conveys the event-selection procedure.

*Figure 5 Clarity: The climatology contours are difficult to distinguish from the zero contour of the anomaly shading. Please use a different color or line style to improve readability.*

**Response:** Thank you for noting this clarity issue. In the revised manuscript, we have adjusted the color and line weight of the climatology contours to clearly distinguish them from the zero-anomaly shading. The updated figure (revised Fig. 5) now provides much improved visual separation and readability.

[Figure]

**Figure 5:** Composite anomalies of 850 hPa geopotential height for different zonal wavenumber planetary wave components during Southern Hemisphere winter (June–August, JJA), overlaid with the long-term climatological mean. (a–c) Composite results for positive MIOD events for zonal wavenumbers 1, 2, and 3, respectively; (d–f) same as (a–c), but for negative MIOD events. Shading indicates geopotential height anomalies, and contours represent the climatological mean.

*Text-Figure Mismatch (Line 347): The text refers to Figure 4b as showing "positive-phase MIOD events," but the figure shows the composite for negative events. Please correct this.*

**Response:** Thank you for noting this oversight. The text has been corrected so that the description of Figure 4b now matches the negative-phase MIOD composite shown in the figure.

*Typographical Errors:*

- *Line 446: The latitude range "50°S-7°S" appears to be a typo and should likely be "50°S-70°S."*
- *Line 555: The sentence beginning "The spatial phase of this cold band..." is redundant and should be revised or removed.*

**Response:** The latitude range has been corrected from "50°S–7°S" to "50°S–70°S,"

and the redundant sentence beginning with "The spatial phase of this cold band…"

has been removed in the revised manuscript.

*Methodological Justification:*

- *E-P Flux Normalization (Figure 6): The non-standard "two-layer normalization approach" requires justification. Explain why this was chosen over standard methods.*

**Response:** Thank you for the comment. A brief clarification has been added to the manuscript: "For clarity, the tropospheric (900-300 hPa) and stratospheric (300-1 hPa)

E-P fluxes are plotted using separate normalization factors to improve the clarity of the plotted vectors since the value in the tropospheric and stratospheric differ substantially in magnitude." in lines **521-523**. This clarification is now included in the text accompanying Figure 7.

- *MIOD Index vs. PC2: Briefly elaborate on why a physically-based box definition is preferable to the mathematically derived PC time series.*

**Response:** Thank you for the comment. The manuscript has been revised to briefly clarify why a physically based SST box index is used instead of the EOF-derived PC2.

The box index provides a more intuitive and stable measure of MIOD variability **and**

**avoids the sensitivity of EOF-based PCs to the choice of analysis period and**

**preprocessing**. In addition, the box index is easier to compute and is directly comparable across datasets and studies, similar to commonly used ENSO indices. This makes it more suitable for identifying individual positive and negative MIOD events and for constructing composites. The revised text now includes this explanation in the section describing the construction of the MIOD index.

It has been explained as "*Using this physically based index rather than PC2 provides a*

*simpler and more intuitive metric for subsequent analyses and avoids the sensitivity of*

*EOF-derived PCs to choices of analysis period and preprocessing*" in lines 324-326 of the revised manuscript.

---

## Author Comment (AC2)

*Comments on "Impact of the Indian Ocean Sea Surface Temperature on the Southern Hemisphere Middle Atmosphere" by Yang et al.*

*This study investigates the impacts of the midlatitude Indian Ocean sea surface temperatures (SSTs) on the Southern Hemisphere middle and upper atmosphere based on the proposed midlatitude Indian Ocean Dipole (MIOD) index. The authors show that positive MIOD events enhance planetary-wave propagation from the Indian Ocean sector, leading to variations in temperature, zonal winds, as well as a strengthening of the residual meridional circulation, while negative MIOD events have relatively weak impacts on the Southern Hemisphere middle and upper atmosphere. The issues tackled in this study are worthwhile and well within the scope of this journal. However, some conclusions are lack of sound verification. It needs major revisions before it is accepted for publication. The following are some specific comments and suggestions:*

*1. Line 38-39: The stratospheric thermal radiation only can not insert significant influences on both tropical and extratropical circulation,it is radiative-chemical-dynamic coupling that is important.*

**Response:** Thank you for the comment. We agree that stratospheric impacts on circulation arise from the combined effects of radiative, chemical, and dynamical processes rather than thermal radiation alone. Accordingly, we have revised the sentence in lines 37–41 to read:

*"Stratospheric processes—including thermal radiation and radiative–chemical–dynamical coupling—have been shown to influence both tropical and extratropical circulation, with further effects on surface temperature (Joshi et al., 2006; Maycock et al., 2013; Shindell, 2001; Solomon et al., 2010; Tandon et al., 2011)."*

This revision clarifies that it is the coupled radiative–chemical–dynamical processes that underpin the stratosphere's influence on the climate system.

*2. Line 104-105: The statement "Yet the atmospheric background conditions during austral winter are more favorable for planetary wave propagation into the stratosphere" needs reference support.*

**Response:** Thank you for the comment. We have added a citation to **Charney and Drazin (1961)**, which demonstrated that planetary waves can propagate vertically only under westerly background flow, thereby providing the theoretical basis for why austral winter conditions favor upward planetary-wave propagation.

3. *Line 164-166: what is the top level of MERRA-2 reanalysis? The WACCM6-SD run at the model top near 140 km. On which model level does the nudging begin to perform?*

**Response:** Thank you for the comment. We have added a clarification in lines 164–170 of the revised manuscript as "*In the SD configuration, meteorological fields are nudged toward MERRA-2 reanalysis every six hours to reduce internal variability and model bias. WACCM6 is nudged toward MERRA-2 below approximately 0.1 hPa (~50–60 km), with a smooth tapering of the relaxation coefficient near the upper boundary of the nudged region. Above this altitude, including the mesosphere and lower thermosphere, the model evolves freely. This setup allows the stratospheric variability to follow the reanalysis while retaining internally generated dynamics in the mesospheric region*".

4. *Line 180: "between 40 and 80 kilometers" >>>"between 40 and 80 km"*

**Response:** Revised.

5. *Line 346: "positive-phase MIOD events" >>> "negative -phase MIOD events"*

**Response:** Revised.

6. *Line 362: what is hgt?*
7. *Line 368: HGT>>hgt*

**Response:** Thank you for the comment. We have clarified in the revised manuscript that *hgt* denotes geopotential height, as introduced in the Data section. In addition, the inconsistent appearance of the uppercase form *HGT* has been corrected, and the notation has been standardized to *hgt* throughout the manuscript to ensure clarity and consistency.

8. *Line 440: Figure 6: Longitude>>Latitude*

**Response:** Revised

9. *Line 510: "ozone deletion" >> "ozone decrease". The depletion generally means destroyed rather the transported.*

**Response:** Thank you for pointing this out. We agree that "ozone depletion" may imply chemical destruction rather than transport-related decreases. We have revised the wording in the manuscript and now use "ozone decrease" to accurately describe the transport-driven changes.

*10. My major concern is related to Section 4. This section presents the results in the mesosphere. It looks strange to put those results in Discussion Section. Are those results are preliminary?*

**Response:** Thank you for raising this important point. Section 4 is not intended to introduce preliminary or additional observational results. Instead, its purpose is to extend the analysis vertically into the mesosphere and to provide a dynamical interpretation of how the MIOD-related stratospheric perturbations documented in Section 3 may project upward. Because gravity-wave processes and MLT variability cannot be directly observed, we combine merged HALOE–SABER temperature data with the free-running mesosphere of SD-WACCM6 to evaluate whether the observed mesospheric patterns are dynamically consistent with those generated internally by the model. To clarify this intent and avoid the impression that Section 4 presents a separate set of results, we have revised the opening paragraph as "*The stratospheric responses described above suggest that MIOD-related perturbations may extend upward into the mesosphere, raising the question of how far the influence of MIOD projects vertically. To investigate the full vertical structure of the atmospheric response, we complement the stratospheric analysis with merged HALOE–SABER temperature observations spanning 10–100 km and SD-WACCM6 simulations. Because the free-running nature of SD-WACCM6 above ~50–60 km allows the mesosphere–lower thermosphere (MLT) variability to evolve independently of the imposed stratospheric state, the comparison between observations and model output provides a basis for examining whether the mesospheric anomalies inferred from observations are dynamically consistent with those that arise internally in the model. This framework enables us to assess potential pathways through which MIOD-related stratospheric perturbations may influence the mesosphere, without presupposing the underlying dynamical mechanism*" in lines 666-677 of the revised manuscript to provide a smoother transition from the stratospheric analysis and updated the title of Section 4 to better reflect its interpretative nature.

*11. Line 540:Figure 9:above 80 km, there is no consistency between the satellite*
*observations and model results. Is it due to nudging approach?*

**Response:** Thank you for the comment. The lack of consistency above ~80 km arises from several factors. First, SD-WACCM6 is nudged toward reanalysis only below approximately 0.1 hPa (~50–60 km), and the mesosphere–lower thermosphere above this level is fully free-running. As a result, the model does not reproduce event- specific variability in the upper mesosphere that is not directly controlled by the imposed lower-atmospheric state. Second, the observational composite (HALOE–

SABER, 1991–2022) and the model composite cover different sampling periods, which may further contribute to differences at altitudes where internally generated variability dominates. We have added a clarification in the revised text as "*However,*

*the midlatitude warming in mesosphere/lower thermosphere region seen in*

*observations is largely absent in the model, and the tropical anomaly remains below 1*

*K and is not statistically significant. This discrepancy between the observations and*

*SD-WACCM6 may indicate that the processes giving rise to the upper-mesospheric*

*and lower-thermospheric response are not fully captured, as SD-WACCM6 is not*

*constrained in the mesosphere. An additional contributing factor may be the non-*

*overlapping portions of the observational record (1991–2022) and the model*

*simulation period used here.*" (Lines 700-706) to make these points explicit.

*12. Line 590-594: The authors stated that "Discrepancies between thermal wind*
*estimates and reanalysis winds are largely attributable to planetary wave*
*breaking". This is not true! various processes may have contributions to those*
*discrepancies.*

**Response:** Thank you for pointing this out. We agree that our original formulation was overly assertive and did not adequately reflect the range of processes that can lead to discrepancies between thermal-wind estimates and reanalysis winds. Our EP- flux and planetary-wave diagnostics suggest that wave forcing is a plausible contributor, but other processes not explicitly analyzed here (e.g., diabatic heating and radiative–chemical tendencies) may also play a role. We have therefore revised the sentence as "*Deviations between thermal-wind estimates and reanalysis winds further*

*point to a dynamical contribution from planetary-wave forcing, although diabatic,*

*radiative, and chemical processes may also play a role*" in lines 748-750 of the revised manuscript to state that planetary-wave forcing likely contributes to the discrepancies, while acknowledging that additional processes may also be important.

*13. Line 597-598: The authors stated that "The influence of the MIOD extends into*
*the mesosphere and lower thermosphere (MLT) through gravity-wave filtering*
*modulated by stratospheric wind perturbations". This statement has no support.*

**Response:** Thank you for the comment. We agree that the original statement overstated the vertical extent and certainty of the mechanism. We have revised the wording to reflect only the supported mesospheric response and to frame the gravitywave contribution more cautiously. The revised sentence now reads: *"In the*

*mesosphere, SD-WACCM6 produces a response that is structurally similar to satellite*

*observations. Within the model, MIOD-related stratospheric wind anomalies*

*modulate gravity-wave filtering and wave-mean flow interactions, leading to coherent*

*mesospheric drag and circulation anomalies. While discrepancies persist, particularly*

*at higher altitudes, these results indicate that gravity-wave filtering provides a*

*physically plausible pathway linking MIOD-related stratospheric disturbances to the*

*mesospheric response"* in lines 751-756 of the revised manuscript.

*14. Line 625-627: "The findings are consistently supported by satellite observations*
*and WACCM6 simulations, lending robustness to the identified SST atmosphere*
*coupling". However, there are no any comparisons between the model results and*
*satellite observations in the stratosphere.*
*15. Line 628-629: "with the Southern Hemisphere atmosphere being more sensitive*
*due to its unique background circulation during winter". There are no any*
*discussions on this statement.*

**Response:** Thank you for these helpful comments. We agree that the original wording overstated both the degree of observational–model consistency and the interpretation of hemispheric sensitivity. We have revised the conclusion to make clear that the consistency between satellite observations and SD-WACCM6 refers specifically to the mesospheric response. We have also removed the statement that the Southern

Hemisphere atmosphere is "more sensitive due to its unique background circulation during winter" and now frame the role of MIOD more generally as a potential additional driver of large-scale atmospheric variability alongside established influences such as ENSO and the QBO. This part has been revised as "*Satellite*

*observations and SD-WACCM6 simulations further indicate that MIOD-related*

*anomalies extend into the Southern Hemisphere mesosphere, with the model*

*suggesting a role for gravity-wave drag modulation in linking stratospheric wind*

*anomalies to the mesospheric response. The MIOD-related atmospheric signal*

*identified here indicates that Indian Ocean SST variability acts as an additional*

*source of large-scale dynamical variability in the Southern Hemisphere,*

*complementing established influences such as ENSO and the QBO, and highlighting a*

*previously underappreciated pathway through which tropical ocean variability affects*

*the middle and upper atmosphere on interannual timescale.*" In lines 773-780 of the revised manuscript.

*16. Line 632-633: "The analysis further suggests that long-term trends in Indian*
*Ocean SST may have contributed to the observed variability in Antarctic ozone*
*depletion and recovery". There are no any discussions on the long term trends of*
*variables. How can you draw this conclusion?*

**Response:** Thank you for pointing this out. We agree that the original sentence introduced an implication regarding long-term SST trends and ozone variability that was not directly analyzed in this study. Since our focus is on the interannual response of the middle and upper atmosphere to MIOD variability, we have removed this statement from the conclusion to avoid overinterpretation.